# Diverse homeostatic and immunomodulatory roles of immune cells in the developing mouse lung at single cell resolution

Racquel Domingo-Gonzalez[1,2†], Fabio Zanini[3,4†], Xibing Che[1,2,5], Min Liu[1,2], Robert C Jones[3], Michael A Swift[6], Stephen R Quake[3,7,8‡*], David N Cornfield[1,2,5‡*], Cristina M Alvira[1,2‡*]

[1]Division of Critical Care Medicine, Department of Pediatrics, Stanford University School of Medicine, Stanford, United States; [2]Center for Excellence in Pulmonary Biology, Stanford University School of Medicine, Stanford, United States; [3]Department of Bioengineering, Stanford University, Stanford, United States; [4]Prince of Wales Clinical School, Lowy Cancer Research Centre, University of New South Wales, Sydney, Australia; [5]Division of Pulmonary, Asthma and Sleep Medicine, Department of Pediatrics, Stanford University School of Medicine, Stanford, United States; [6]Department of Chemical and Systems Biology, Stanford University, Stanford, United States; [7]Chan Zuckerberg Biohub, San Francisco, United States; [8]Department of Applied Physics, Stanford University, Stanford, United States

**\*For correspondence:**
steve@quake-lab.org (SRQ);
cornfield@stanford.edu (DNC);
calvira@stanford.edu (CMA)

[†]These authors contributed equally to this work
[‡]These authors also contributed equally to this work

**Competing interests:** The authors declare that no competing interests exist.

**Abstract** At birth, the lungs rapidly transition from a pathogen-free, hypoxic environment to a pathogen-rich, rhythmically distended air-liquid interface. Although many studies have focused on the adult lung, the perinatal lung remains unexplored. Here, we present an atlas of the murine lung immune compartment during early postnatal development. We show that the late embryonic lung is dominated by specialized proliferative macrophages with a surprising physical interaction with the developing vasculature. These macrophages disappear after birth and are replaced by a dynamic mixture of macrophage subtypes, dendritic cells, granulocytes, and lymphocytes. Detailed characterization of macrophage diversity revealed an orchestration of distinct subpopulations across postnatal development to fill context-specific functions in tissue remodeling, angiogenesis, and immunity. These data both broaden the putative roles for immune cells in the developing lung and provide a framework for understanding how external insults alter immune cell phenotype during a period of rapid lung growth and heightened vulnerability.

## Introduction

Prior to birth, the lung is maintained in a fluid-filled, immune-privileged, hypoxic environment. Upon the infant's first breath, the lung rapidly transitions to an oxygen rich environment (*Koos and Rajaee, 2014*), subjected to the mechanical forces of spontaneous ventilation, and exposed to diverse pathogens present in the external environment (*Wirtz and Dobbs, 2000*). The immune system is essential for lung homeostasis, wound-healing and response to pathogens (*Lloyd and Marsland, 2017*). Although the development of the murine immune system begins during early embryogenesis, little is known regarding how the dynamic physiologic changes at birth alter the lung immune cell

landscape, and whether specific immune cell subpopulations influence lung growth and remodeling in addition to serving established immunomodulatory functions.

Immune cells play a central role in the development of many organs, including the kidney (*Munro and Hughes, 2017*), brain (*Lenz and Nelson, 2018*) and mammary gland (*Coussens and Pollard, 2011*). In highly vascularized organs, macrophages localize to the tips of vascular sprouts to enhance vascular network complexity (*Rymo et al., 2011*), promote angiogenesis (*Riabov et al., 2014*), and regulate vascular patterning (*Leid et al., 2016*). Although proximal lung branching occurs during early gestation, the development of distal airspaces begins only just before birth, and continues postnatally during alveolarization, the final stage of lung development characterized by rapid lung parenchymal and vascular growth (*Rodríguez-Castillo et al., 2018*). Disruption of late lung development contributes to pediatric lung diseases such as bronchopulmonary dysplasia (BPD), a disease of arrested lung development observed in premature infants, and pulmonary hypertension. Whether temporal regulation of specific immune populations informs lung immune function or the significant pulmonary parenchymal and vascular growth and remodeling occurring during early postnatal life remains unknown.

The prevailing notion is that the neonatal immune compartment is immature (*Restori et al., 2018*). Attenuated innate immunity (*Sadeghi et al., 2007*), poor immune-stimulatory function of antigen presenting cells (*De Kleer et al., 2014*), and skewed adaptive immune responses may underlie the heightened susceptibility of infants to viral and bacterial infections (*Restori et al., 2018*). Although the neonatal immune system can be induced to manifest adult-like responses under certain conditions (*Pertmer et al., 2001*), this type-2-skewed immune environment likely facilitates immune surveillance and metabolic and tissue homeostasis (*Lloyd and Snelgrove, 2018*). Given that lung development continues as the immune cell landscape is rapidly evolving, identifying how birth-related changes in lung physiology may influence lung immune cell maturation and function has not been addressed.

In this report, we combined single cell transcriptomics (scRNA-Seq) with fluorescent multiplexed in situ hybridization (FISH) and flow cytometry to characterize changes in composition, localization, and function of immune cells in the murine lung from just before birth through the first three weeks of postnatal life. At birth, immune cell heterogeneity increased dramatically from an embryonic landscape dominated by immature, proliferative macrophages to a complex landscape comprised of multiple types of macrophages, dendritic cells, granulocytes, and lymphocytes. Dynamic changes in macrophage heterogeneity were particularly striking, both transcriptionally and spatially, including the presence of embryonic macrophages encircling developing vessels prior to birth. After birth, these embryonic macrophages disappear, and numerous unique macrophage populations emerge, each exhibiting unique gene signatures suggesting specific roles in immunosuppression, pathogen surveillance, angiogenesis, and tissue remodeling. Multiple populations of dendritic cells, basophils, mast cells and neutrophils are also present in the postnatal lung, expressing genes important for rapid pathogen response. In contrast, although lymphocytes increase in abundance across three weeks, they remain functionally immature and skewed toward type-2 immunity. Taken together, our data demonstrate a previously underappreciated plasticity of immune cells in the perinatal and neonatal lung suggesting unique and essential roles in regulating immune function and lung structure.

## Results

### Diversity of the lung immune landscape increases dramatically after birth

To comprehensively define the lung immune landscape at birth, we isolated whole lungs from C57BL/6 (B6) mice at four stages of perinatal development: the early saccular (E18.5), late saccular (P1) early alveolar (P7) and late alveolar stages (P21) (*Figure 1A*), and quantified gene expression by scRNA-Seq. Lung tissue was isolated, the pulmonary vasculature perfused to remove circulating immune cells, and the tissue digested using an in-house optimized protocol to ensure maximal cell viability as published protocols (*Nguyen et al., 2018*; *Jungblut et al., 2009*) induced high amounts of cell death in the embryonic and early postnatal lung (*Figure 1—figure supplement 1*). Live CD45 + cells were sorted by FACS, processed by Smart-seq2 and sequenced on Illumina NovaSeq 6000 (*Figure 1A*). Gene expression was computed as previously described (*Zanini et al., 2018*) and 4,052

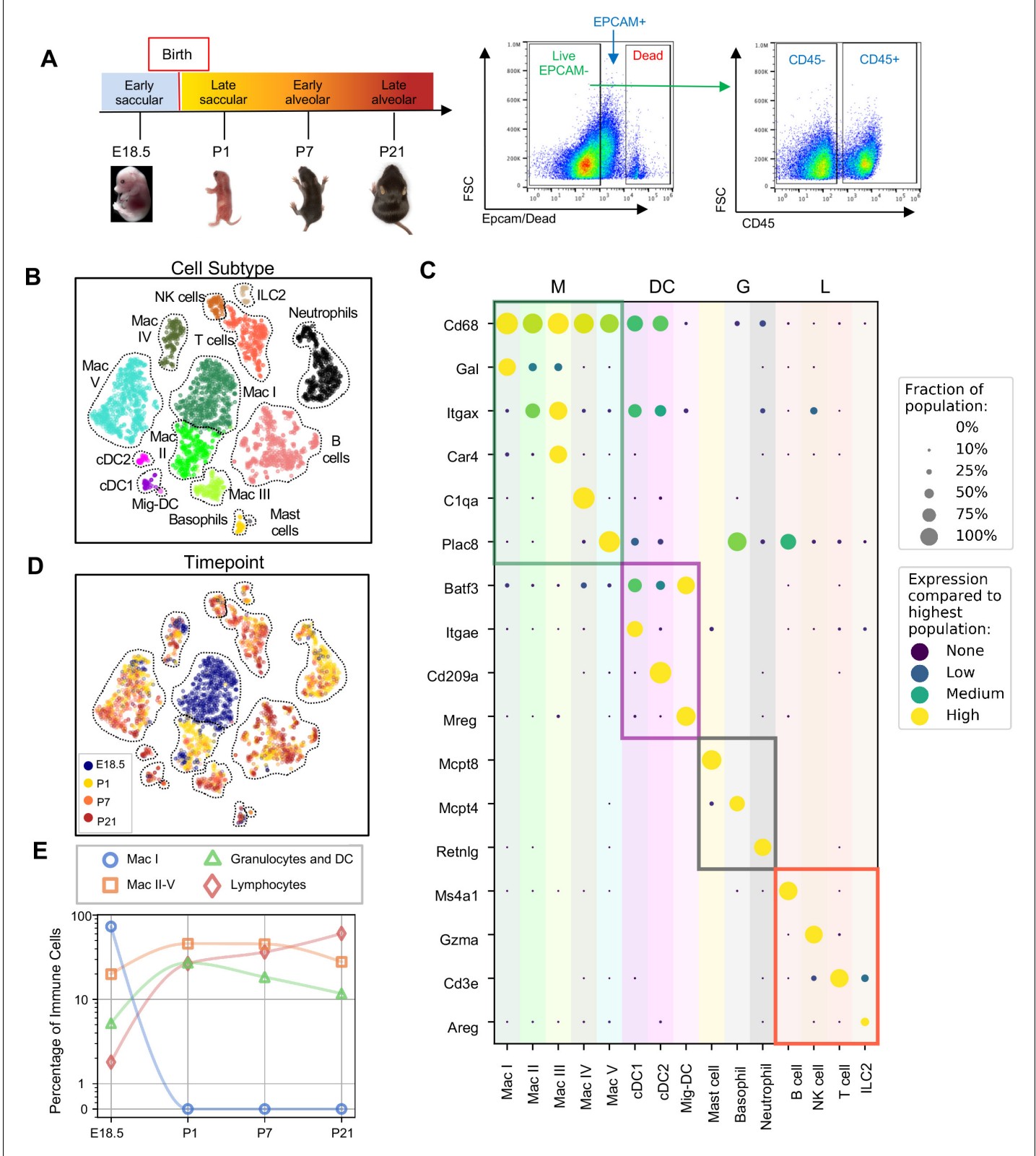

**Figure 1.** Diversity of the lung immune landscape increases dramatically after birth. (A) Overview of the experimental design including the four timepoints (E18.5, P1, P7, P21) corresponding to key stages in late lung development. Lungs were isolated, perfused, and digested and immune cells isolated by fluorescence activated cell sorting (FACS) for the dead-stain-, EPCAM-, CD45+ population. (B) t-Distributed Stochastic Neighbor Embedding (t-SNE) and unsupervised clustering of over 4000 immune cells identifies fifteen distinct populations. (C) Dot plot showing level of

*Figure 1 continued on next page*

*Figure 1 continued*

expression (purple to yellow), and fraction of the population expressing the particular gene (dot size) for distinguishing genes expressed by the Leiden clusters broadly separated into myeloid (M), dendritic cell (DC) granulocyte (G) and lymphocyte (L) populations. (D) t-SNE of immune cell clusters identifying developmental timepoint of cell origin with E18.5 (blue), P1 (yellow), P7 (orange) and P21 (red). (E) Quantification of the abundance of specific immune subpopulations in the lung at each developmental timepoint expressed on a log (*Rodríguez-Castillo et al., 2018*) scale as percentage of total immune cells.

The online version of this article includes the following figure supplement(s) for figure 1:

**Figure supplement 1.** Optimization of lung tissue digestion.

**Figure supplement 2.** Determination of variation between mice.

cells from eight mice, 1 female and one male for each time point were analyzed, with an average of 1.03 million mapped read pairs and ~4000 genes per cell (see Materials and methods). To quantify whether the different mice contributed spurious variation to the data, a distribution level approach was chosen. For each cell type and time point, 100 pairs of cells from either the same mouse or different mice were chosen and the distance in t-SNE space calculated. The cumulative distributions for those pairs were subsequently plotted to check whether pairs from different animals had a significantly longer distance than cells from the same mouse. We found no difference in the cumulative distributions (as also evident, on a qualitative level, by observation of the embeddings), indicating the absence of significant variation between the mice at each timepoint (*Figure 1—figure supplement 2*).

Fifteen cell clusters were identified via Leiden community detection (*Traag et al., 2019*) and verified by t-distributed stochastic neighbor embedding (t-SNE) (*van der Maaten and Hinton, 2008*; *Figure 1B*). Myeloid cells separated into eleven clusters, including five distinct macrophage/monocyte subpopulations with shared expression of *Cd68,* and distinguished by expression of *Gal* (Mac I), *Itgax* (Mac II), *Car4* and *Itgax* (Mac III), *C1qa* (Mac IV), or *Plac8* (Mac V). Dendritic cells (DCs) separated into three clusters, all expressing some amount of *Batf3*, but distinguished by the expression of *Itgae* (cDC1), *Cd209a* (cDC2), or *Mreg* (mig-DC). We also identified mast cells (expressing *Mcpt4*), basophils (*Mcpt8*), and neutrophils (*Retnlg*). Four lymphoid clusters were found, consisting of B cells (expressing *Ms4a1*), T cells (*Cd3e*), natural killer (NK) cells (*Gzma*), and group two innate lymphoid cells (ILC2) (*Areg*) (*Figure 1C*).

We next assessed cluster distribution across time (*Figure 1D and E*). Mac I cells dominate the late embryonic lung, with fewer macrophages scattered among clusters II, IV and V and an even smaller number of granulocytes and lymphocytes. After birth, immune cell heterogeneity increased explosively, concomitant with the disappearance of Mac I. Granulocyte abundance peaked just after birth and lymphocyte abundance increased progressively.

## Expression of *Dab2* and *Plac8* broadly separates macrophages and monocytes

Clusters Mac I-V exhibited the most striking heterogeneity, so we analyzed their transcriptomes and spatial distribution in detail. All five clusters shared high expression of *Cd68*, indicative of macrophages or monocytes (m/m) (*Figure 2A*). At E18.5, Mac I comprised ~80% of m/m cells and Mac II, IV, and V each comprised 5–10% while Mac III was almost absent (*Figure 2B and C*). After birth, Mac I disappeared while Mac II abundance peaked to 35% of the total before decreasing again and disappearing by P21. Mac III and Mac V abundance increased steadily with Mac V being most abundant at all postnatal timepoints. The Mac IV population was relatively stable over time at ~10% of the total.

The Mac I-V clusters broadly separated into two groups based upon expression of the disabled two gene (*Dab2*), which regulates macrophage polarization (*Adamson et al., 2016*) and the placenta-specific eight gene (*Plac8*), which is related to bacterial immunity (*Ledford et al., 2007*). Mac I-IV cells expressed *Dab2* but not *Plac8* while Mac V showed the opposite pattern (*Figure 2D*). We confirmed by multiplexed FISH that all *Cd68*[+] cells in the lung expressed either *Dab2* or *Plac8* at both E18.5 and P7 (*Figure 2E* and *Figure 2—figure supplement 1*). Quantification in situ identified 88.9 ± 8.4% of E18.5 and 65.8 ± 8.2% of P7 *Cd68*[+] cells as *Dab2*+ with the remainder *Plac8*+. Of note, *Dab2* and *Plac8* expression was not consistent with previously reported markers of

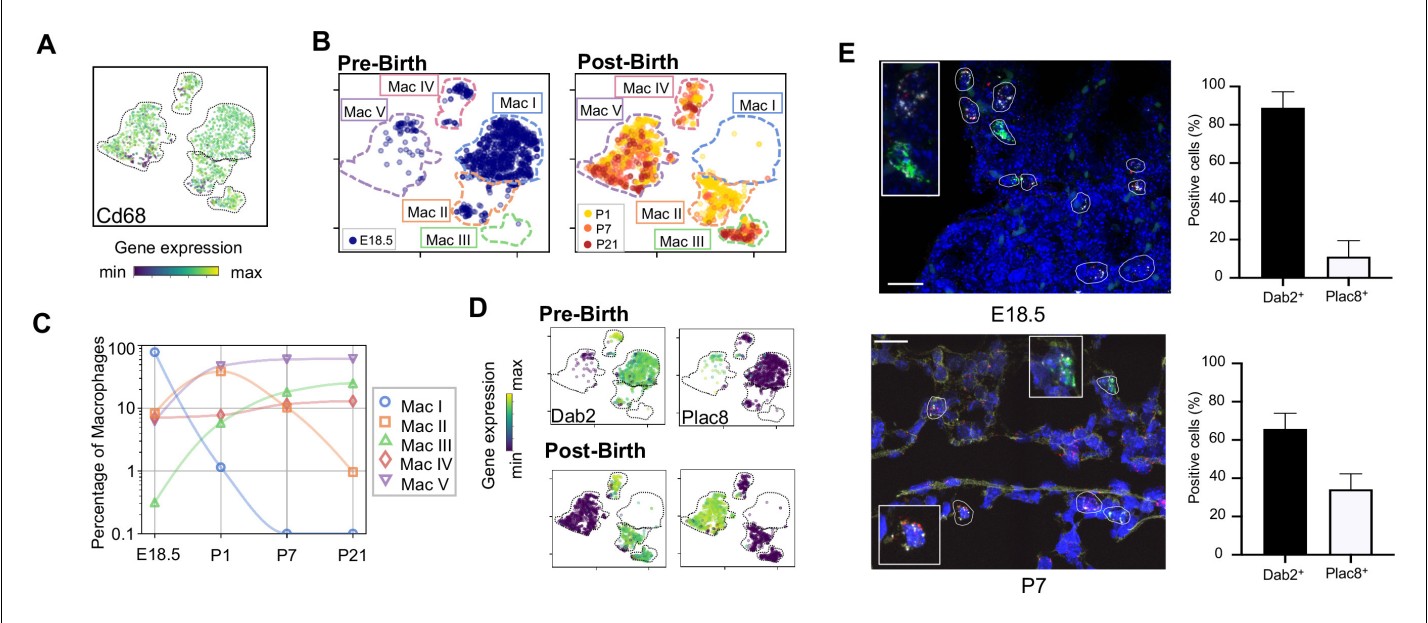

**Figure 2.** Macrophage populations present before and after birth broadly separate into two populations based on expression of *Dab2* and *Plac8*. (A) t-SNE plot depicting *Cd68* expression in the five macrophage populations. (B) Separate embeddings for prenatal versus postnatal macrophages, identifying developmental timepoint of cell origin with E18.5 (blue), P1 (yellow), P7 (orange) and P21 (red). (C) Quantification of the abundance of each macrophage subpopulation at each developmental timepoint expressed on a log (*Rodríguez-Castillo et al., 2018*) scale as percentage of total macrophages. (D) t-SNE plots depicting expression of *Dab2* and *Plac8* within the macrophages present pre- and post-birth. (E) Multiplexed in situ hybridization to detect gene expression of *Cd68* (yellow), *Dab2* (red), and *Plac8* (green) in lung tissue from mice at E18.5 and P7. Quantification of *Dab2*+ and *Plac8*+ cells with data = mean ± SD for n = 4–15 images from three separate FISH experiments. Calibration bar = 20 μm. For all t-SNE embeddings, the color scale is a logarithmic scale with a pseudocount of 0.1 counts per million, normalized to the highest expressing cell. Source files of all fluorescent micrographs used for the quantitative analysis are available in the *Figure 2—source data 1*.

The online version of this article includes the following source data and figure supplement(s) for figure 2:

**Source data 1.** Source files for quantification of Dab2+ and Plac8+ Cd68+ cells.
**Figure supplement 1.** Multiplex in situ hybridization to detect Dab2 and Plac8 expressing macrophages.
**Figure supplement 2.** Lineage-defining genes are diffusely expressed across macrophage populations.

macrophage lineages derived from yolk sac and fetal liver (*Tan and Krasnow, 2016*; *Guilliams et al., 2013*; *Figure 2—figure supplement 2*).

## Embryonic macrophages are proliferative and encircle developing vessels prior to birth

Mac I cells are the predominant immune population at E18.5, hence we aimed to understand their function and localization. Differentially expressed genes (DEGs) in Mac I included the proliferation markers *Mki67* and *Mcm5* (*Figure 3A*). Across macrophages and monocytes, proliferation decreased from 60% of cells at E18.5 to only 10% by P21. Most proliferating cells were Mac I prior to birth and distributed across Mac II-V after birth (*Figure 3—figure supplement 1*). These data are consistent with prior reports of 'bursts' of proliferation after recruitment of macrophages into embryonic tissues, followed by low-level self-renewal by adulthood (*Kierdorf et al., 2015*).

Localization of Mac I cells within the E18.5 lung revealed *Cd68+* cells scattered throughout the lung parenchyma but also, surprisingly, forming almost complete rings around blood vessels of 20–30 μm in diameter found adjacent to large, conducting airways (*Figure 3B*). In contrast, Mac I cells were not found encircling small airways (*Figure 3C*). Quantification of the percentage of *Cd68*+ cells located around vessels versus in the lung parenchyma identified 28.4 ± 14.2% of *Cd68*+ cells surrounding small vessels (*Figure 3D*). Many of the perivascular macrophages expressed *Mki67* (*Figure 3E*), and quantification of *Mki67*+ macrophages identified 55.2 ± 9.3% of proliferative perivascular *Cd68*+ cells, and a similar number of proliferative parenchymal *Cd68*+ cells (51.1 ± 16.8%). Perivascular macrophages were found on in situ analysis to express *Dab2* but not *Plac8* (*Figure 3F*).

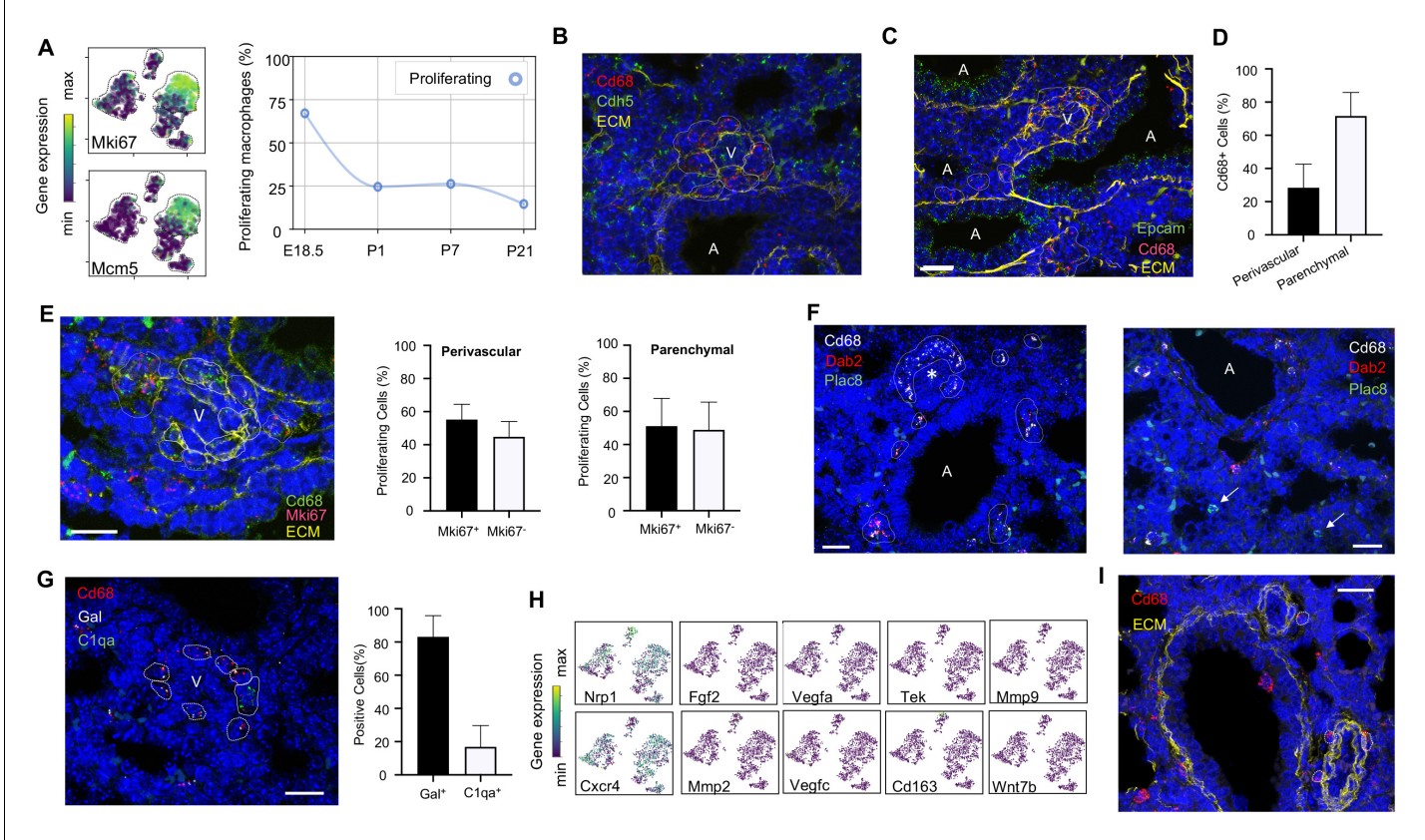

**Figure 3.** Embryonic macrophages encircle developing blood vessels prior to birth. (A) t-SNE plots depicting expression of *Mki67* and *Mcm5* in the macrophage clusters with low expression in purple and high expression in yellow, with quantification of proliferating macrophages at each timepoint. In situ hybridization at E18.5 to detect: (B) *Cd68* (red) and *Cdh5* (green) with dotted lines identifying macrophages in circles around small vessels; (C) *Epcam* (green), *Cd68* (red), and extracellular matrix (ECM, yellow), with white dotted circles identifying *Cd68*+ cells or groups of cells; (D) Quantification of the number of *Cd68*+ cells around vessels versus in lung parenchyma, with data = mean ± SD in n = 14 images, from five independent FISH experiments. (E) In situ hybridization at E18.5 to detect *Mki67* (red), *Cd68* (green), and ECM (yellow), with white dotted circles identifying *Cd68*+*Mki67*+ cells, and solid circles identify *Cd68*+ *Mki67*- cells. Quantification of the number of perivascular and parenchymal *Mki67*+ and *Mki67*- *Cd68*+ cells in n = 10 images with data = mean ± SD from two independent FISH experiments. (F) In situ hybridization at E18.5 to detect *Cd68* (white) *Dab2* (red), and *Plac8* (green) with white dotted circles identifying *Cd68*+*Dab2*+ macrophages in the left panel, and arrows identifying *Plac8*+ cells in a separate area from the same slide (right panel). (G) In situ hybridization at E18.5 to detect *Cd68* (red), *Gal (white)*, and *C1qa* (green), with quantification of *Gal*+ and *C1qa*+ *Cd68*+ cells with data = mean ± SD from n = 12 images from three independent FISH experiments. (H) t-SNE plots of genes previously associated with a perivascular macrophage phenotype. (I) In situ hybridization of lung at P1 to detect *Cd68* (red) and ECM (yellow), with white dotted circles identifying isolated macrophages around blood vessels. In each micrograph, calibration bar = 20 µm and 'V' denotes 'vessel and 'A' 'airway'. For all t-SNE embeddings, the color scale is a logarithmic scale with a pseudocount of 0.1 counts per million, normalized to the highest expressing cell. Source files of all fluorescent micrographs used for the quantitative analysis are available in the *Figure 3—source data 1*, *Figure 3—source data 2*, and *Figure 3—source data 3*.

The online version of this article includes the following source data and figure supplement(s) for figure 3:

**Source data 1.** Source files for quantification of perivascular and parenchymal Cd68+ cells at E18.5.
**Source data 2.** Source files for quantification of Mki67+ Cd68+ cells at E18.5.
**Source data 3.** Source files for quantification of Gal+ and C1qa+ perivascular Cd68+ cells at E18.5.
**Figure supplement 1.** Distribution of proliferating cells among macrophage/monocyte clusters.

Given that *Dab2*+ cells at E18.5 include Mac IV cells, we aimed to distinguish these from the Mac I cells by simultaneously detecting the expression of *Gal* and *C1qa*, which were determined to be specific markers (see below). These studies demonstrated that the majority of *Cd68*+ macrophages surrounding the vessels were *Gal*+ (83.1 ± 12.8%) (*Figure 3G*), indicating a predominance of Mac I and a small number of Mac IV cells among the perivascular macrophage population. We then asked whether any of our Mac I-V clusters are related to previously reported perivascular macrophages

that promote vascular remodeling in the developing hindbrain and retina (*Pucci et al., 2009*). The Mac I cluster expressed *Cxcr4* and *Nrp1* but failed to express many other genes characteristic of these previously reported macrophages, suggesting a distinct phenotype (*Figure 3H*). Moreover, concentric perivascular macrophages were not observed at P1, consistent with a function specific to Mac I prior to birth (*Figure 3I*). Taken together, these data suggest that within the embryonic lung, Mac I macrophages are highly proliferative and localize to small vessels, suggesting a potential role in pulmonary vascular growth or remodeling.

## Distinct transcriptional profiles and spatial distribution suggest specific physiologic functions for discrete macrophage populations

Macrophage and monocyte heterogeneity increased rapidly after birth. To characterize this expanding diversity we computed DEGs and performed pathway analysis for each of the Mac I-V clusters (*Figure 4A*, *Figure 4—figure supplement 1*, and *Tables 1–5*). The most enriched pathways for Mac I were pathways associated with mitosis, cell cycle and DNA replication, consistent with the increased rate of proliferating cells within this cluster. Beyond proliferation, examination of the top DEGs identified genes associated with glycolysis, reflective of the hypoxic fetal environment compared to postnatal air-breathing life. Mac I-specific DEGs also included *Crispld2,* a glucocorticoid-regulated gene previously thought to be restricted primarily to the lung mesenchyme that promotes lung branching (*Oyewumi et al., 2003*). *Crispld2* haploinsufficient mice exhibit impaired alveolarization and disorganized elastin deposition (*Lan et al., 2009*). Mac I cells also expressed *Spint1*, encoding hepatocyte growth factor activator inhibitor type 1 (HAI-1), a membrane bound serine proteinase inhibitor and regulator of angiogenesis (*Figure 4A* and *Table 1*). Loss of *Spint1,* results in a complete failure of placental vascularization and embryonic lethality at E10 that appears to result from a loss of basement membrane integrity (*Fan et al., 2007*). The most specific marker for Mac I was *Gal*, encoding galanin, a regulatory peptide that harbors both pro- and anti-inflammatory functions (*Lang et al., 2015*), promotes an anti-thrombotic phenotype in endocardial EC (*Tyrrell et al., 2017*), and regulates growth and self-renewal of embryonic stem cells (*Tarasov et al., 2002*). Galanin also inhibits inflammatory and histamine-induced vascular permeability in a number of experimental models (*Ji et al., 1995*; *Jancsó et al., 2000*; *Holmberg et al., 2005*), and functions as a vasoconstrictor, limiting blood flow in the cutaneous microcirculation (*Schmidhuber et al., 2007*). Localization identified Mac I cells throughout the lung parenchyma in addition to those found encircling small vessels (*Figure 4B*).

The Mac II cluster rapidly appeared after birth and expressed a gene signature suggesting a putative role in immune regulation and tissue remodeling. Pathway analysis identified top pathways regulating neutrophil homeostasis, endocytosis, apoptosis, and motility (*Figure 4—figure supplement 1*). Examination of the top DEGs identified shared expression of the chemokine receptors *Ccr2* and *Ccr5* with Mac I, molecules important for immune cell migration and localization. Also similar to Mac I and III, Mac II cells expressed genes associated with matrix remodeling and angiogenesis (*Fn1, Axl, and Tgm2*) (*Senger and Davis, 2011*; *Tanaka and Siemann, 2019*; *Figure 4A* and *Table 2*). We also examined expression patterns of secreted factors that regulate key signaling pathways important for lung epithelial progenitor cell function and differentiation including the Wnt, BMP, Tgf-β, Fgf and Shh pathways (*Morrisey and Hogan, 2010*; *Figure 4—figure supplement 2*). In addition to *Tgfb1* and *Tgfb2*, Mac II cells expressed *Ntn1* and *Ntn4*, molecules that modulate the response of the lung endoderm to FGFs, regulating epithelial bud outgrowth and shape (*Liu et al., 2004*), and *Hhip*, a Hedgehog binding protein that also modulates FGF signaling (*Chuang et al., 2003*). *Hhip* null mice exhibit defective lung branching and haploinsufficient mice develop spontaneous emphysema (*Lao et al., 2016*). The Mac II cluster also shared genes with Mac III important for regulating inflammation, including genes that promote intracellular killing of pathogens (*Acp5* and *Mpeg1*) (*Benard et al., 2015*), but also genes that suppress inflammation (*Il1rn and Dapk1*) (*Iannitti et al., 2016*; *Lai and Chen, 2014*). A subpopulation within Mac II shared high expression of major histocompatibility complex (MHC) class II genes (*H2-Ab1*, *H2-Eb1*, *Cd74*) with a subset of Mac IV (*Figure 4—figure supplement 3*) suggesting a role in antigen presentation. Overall, the transient presence of Mac II together with its transcriptional signature suggests a dual role in tissue remodeling and fine tuning of immune response required immediately after birth. Studies to localize *Itgax +Car4-* Mac II cells in situ at P1 identified them in the distal lung parenchyma, mixed with scattered

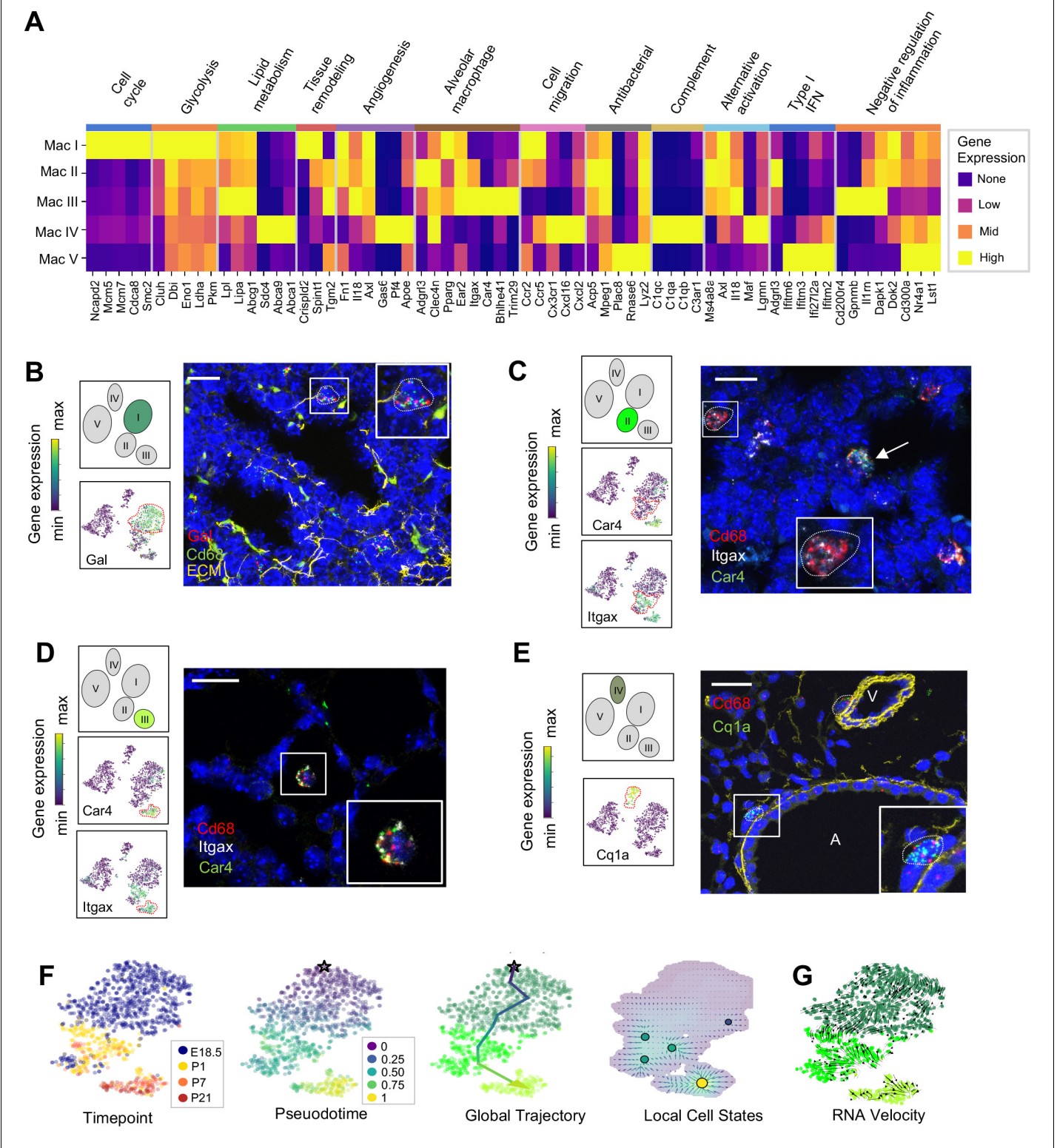

**Figure 4.** Distinct transcriptional profiles and spatial distribution suggest specific physiologic functions for discrete macrophage populations. (A) Heatmap of select differentially expressed genes within enriched pathways illustrated. The color scale is logarithmic with a pseudocount of 0.1 counts per million, normalized to the highest expressing population for each gene, depicting 0 (none), 33 (low), 67 (mid), and 100% (high) expression compared to the highest expressing population. (B) t-SNE plots demonstrating high expression of *Gal* in Mac I cells, and in situ hybridization at E18.5 to detect Mac I cells that co-express *Gal* (red) and *Cd68* (green). (C) t-SNE plots demonstrating high expression of *Itgax* but not *Car4* in Mac II cells,

*Figure 4 continued on next page*

*Figure 4 continued*

and in situ hybridization to detect Mac II cells at P1 expressing *Itgax*, and *Cd68* but not *Car4* (dotted line), and additional Mac III cells in the same image co-expressing *Itgax*, *Cd68*, and *Car4* (arrow). (D) t-SNE plots demonstrating high expression of *Itgax* and *Car4* in Mac III cells, and in situ hybridization detecting Mac III cells at P7 expressing *Itgax*, *Car4* and *Cd68* now located within alveoli. (E) t-SNE plot demonstrating high expression of *C1qa* in Mac IV cells, and in situ hybridization to detect Mac IV cells expressing *C1qa (green)* and *Cd68(red)* at P7, with ECM marked in yellow, localizing Mac IV cells abutting vessels and large airways. Calibration bar = 20 μm for all panels. For all t-SNE embeddings, the color scale is a logarithmic scale with a pseudocount of 0.1 counts per million, normalized to the highest expressing cell. (F) t-SNE plot showing a developmental gradient across Mac I-III. Pseudotime ordering of the cells identified a global trajectory from the starting cell (star) in Mac I to Mac III, and local cell states revealing multiple areas of local attraction within the Mac II and Mac III clusters. (G) RNA velocity demonstrating local group velocity from Mac I to Mac II and continuation of the vectors through Mac III.

The online version of this article includes the following figure supplement(s) for figure 4:

**Figure supplement 1.** Pathway analysis of differentially expressed genes in the macrophage populations.
**Figure supplement 2.** Select genes involved in pathways regulating lung progenitor cells are expressed by Mac II and Mac IV.
**Figure supplement 3.** Expression of select genes in Mac IV cluster.
**Figure supplement 4.** Multiplex in situ hybridization to detect specific macrophage clusters.

Mac III cells which were located closer to the alveolar wall (*Figure 4C* and *Figure 4—figure supplement 4*).

The Mac III cluster uniquely expressed alveolar macrophage signature genes (*Car4*, *Bhlhe41*, *Trim29*) (*Figure 4A*, *Table 3*). Top pathways included the inflammatory response (*Figure 4—figure supplement 1*), including numerous genes that constrain inflammation (*Cd200r4*, *Gpnmb*, *Il1rn*) (*Ripoll et al., 2007*; *Zhang et al., 2004*), and lipid catabolic processes, including the genes *Lpl*, *Lipa*, and *Abcg1*, indicative of their essential role in surfactant catabolism (*Eckert et al., 1983*; *Baker et al., 2010*). Tissue remodeling was also identified as a top pathway, including *Ctsk*, a cysteine protease important for the maintenance of airway structural integrity (*Zhang et al., 2011*), *Serpine1*, a serine protease that can either inhibit (*Wu et al., 2015*) or promote angiogenesis (*Bajou et al., 2014*), and *Il18*, a pro-inflammatory cytokine that drives both pulmonary vascular and parenchymal remodeling (*Rodriguez-Menocal et al., 2014*; *Kang et al., 2012*). In situ imaging of *Itgax+ Car4+* cells demonstrated that they move from the distal lung parenchyma at P1 to the alveolar lumen by P7 (*Figure 4D*), confirming their identification as alveolar macrophages (AMs).

Mac IV uniquely expressed numerous proinflammatory genes (*Pandya and Wilkes, 2014*; *Figure 4A*), in contrast to the balanced inflammatory signature of Mac II and AMs. Top enriched pathways included leukocyte chemotaxis, inflammatory response, cytokine production and regulation of SMC proliferation (*Figure 4—figure supplement 1*). These included genes in the classical complement pathway (*C1qa*, *C1qb*, *C1qc*, *C3ar1*) and CCR2 ligand *Ccl12* (*Table 4*), suggesting a role in the localization of CCR2 expressing monocytes (*Griffith et al., 2014*). The Mac IV cells also expressed *Cxcl16*, an IFNγ regulated chemokine that promotes T cell recruitment through CXCR6 (*Matsumura et al., 2008*). In addition to these immunomodulatory functions, the Mav IV also expressed factors with established roles in lung parenchymal and vascular homeostasis. These included *Igf1*, a potent mitogen and survival factor for vascular cells, recently shown to preserve lung growth and prevent pathologic vascular remodeling in an experimental model of BPD (*Seedorf et al., 2020*), *Ang*, a potent vasoconstrictor, *Pf4*, an angiostatic factor that inhibits EC proliferation (*Maione et al., 1990*), and ApoE, a lipoprotein that modulates endothelial NO production (*Sacre et al., 2003*). Further, Mac IV expressed secreted factors important from lung epithelial progenitors distinct from those expressed by Mac II, including *Fgf10*, a growth factor required for early lung branching morphogenesis that also serves to promote epithelial progenitor cell proliferation and mesenchymal cell differentiation (*Sekine et al., 1999*; *Ramasamy et al., 2007*), *Bmp4*, and *Wnt2* and *Wnt5a* (*Figure 4—figure supplement 2*). Mac IV highly expressed *Mrc1* (CD206) (*Figure 4—figure supplement 3*), but within the Mac IV cluster there were a group of cells with lower *Mrc1* and high MHC II gene expression (*H2-Ab1*, *H2-Eb1*, and *Cd74*). These transcriptional differences within Mac IV are similar to two recently reported interstitial macrophages (IMs) in adult lung that can be differentiated by expression of Mrc1 and MHC II genes (*Schyns et al., 2019*). However, the other genes reported to distinguish those two clusters (e.g *Cx3cr1*, *Mertk*, *Cclr2*, *Cd68*) (*Figure 4—figure supplement 3*) were diffusely expressed throughout the Mac IV cluster. In situ imaging to localize *C1qa+Cd68+* cells in the postnatal lung found Mac IV cells remained abutting small

**Table 1.** Top 35 differentially expressed genes in the Mac I cluster.
ScRNA-Seq was performed on live CD45+ lung cells isolated from E18.5, P1, P7 and P21 B6 pups. Top 35 upregulated genes are shown for the Mac I cluser; n = 2 mice.

**Mac I cluster top 35 differentially expressed genes**

| Gene name | Statistic | P value | Log2 fold change |
|---|---|---|---|
| Gal | 0.6133 | 5.42E-53 | 3.6207 |
| Crispld2 | 0.5 | 1.89E-34 | 3.6018 |
| Ncapd2 | 0.4567 | 1.40E-28 | 3.3349 |
| Pclaf | 0.49 | 4.88E-33 | 3.1388 |
| Mcm5 | 0.5067 | 2.07E-35 | 2.9792 |
| Cdca8 | 0.44 | 1.74E-26 | 2.8548 |
| Lig1 | 0.45 | 9.89E-28 | 2.7655 |
| AI506816 | 0.5233 | 7.05E-38 | 2.7046 |
| Asf1b | 0.4433 | 6.73E-27 | 2.5712 |
| Smc2 | 0.4367 | 4.44E-26 | 2.4042 |
| Fkbp5 | 0.5533 | 1.42E-42 | 2.3039 |
| Mcm7 | 0.44 | 1.74E-26 | 2.1166 |
| Cks1b | 0.45 | 9.89E-28 | 2.0842 |
| A930007I19Rik | 0.46 | 5.23E-29 | 2.0187 |
| Psat1 | 0.52 | 2.24E-37 | 2.0158 |
| Pfkl | 0.4467 | 2.59E-27 | 1.9417 |
| Ezh2 | 0.4433 | 6.73E-27 | 1.8453 |
| Dkc1 | 0.4733 | 9.21E-31 | 1.8027 |
| Cluh | 0.44 | 1.74E-26 | 1.7959 |
| Scd2 | 0.44 | 1.74E-26 | 1.7455 |
| Ruvbl2 | 0.4433 | 6.73E-27 | 1.7334 |
| Mif | 0.4933 | 1.67E-33 | 1.6941 |
| Tmem273 | 0.44 | 1.74E-26 | 1.628 |
| Stmn1 | 0.4433 | 6.73E-27 | 1.5905 |
| Spint1 | 0.5267 | 2.20E-38 | 1.5781 |
| Tpi1 | 0.5033 | 6.29E-35 | 1.562 |
| C230062I16Rik | 0.45 | 9.89E-28 | 1.5462 |
| Nt5dc2 | 0.4467 | 2.59E-27 | 1.5376 |
| Ran | 0.4567 | 1.40E-28 | 1.5078 |
| Trf | 0.5933 | 2.32E-49 | 1.4397 |
| Gcnt1 | 0.44 | 1.74E-26 | 1.4095 |
| Abcd2 | 0.46 | 5.23E-29 | 1.3537 |
| Anp32b | 0.47 | 2.56E-30 | 1.3418 |
| Dbi | 0.5167 | 7.04E-37 | 1.2018 |
| Atp5g3 | 0.52 | 2.24E-37 | 1.1808 |

blood vessels as well as the abluminal side of large airways (*Figure 4E*). The characteristic location and the expression of numerous genes important for leukocyte recruitment and pathogen defense suggest that Mac IV may serve as patrollers, playing a key role in immune-surveillance, innate pathogen defense, and antigen presentation, and well as informing lung vascular and airway structure or remodeling.

**Table 2.** Top 35 differentially expressed genes in the Mac II cluster.
ScRNA-Seq was performed on live CD45+ lung cells isolated from E18.5, P1, P7 and P21 B6 pups. Top 35 upregulated genes are shown for the Mac II cluster; n = 2 mice.

**Mac II cluster top 35 differentially expressed genes**

| Gene name | Statistic | P value | Log2 fold change |
|---|---|---|---|
| Adgrl3 | 0.5713 | 8.85E-42 | 3.8593 |
| Adarb1 | 0.4216 | 5.92E-23 | 2.6952 |
| Ms4a8a | 0.4528 | 2.11E-26 | 2.4485 |
| Nav2 | 0.4751 | 4.97E-29 | 2.1319 |
| Tmcc3 | 0.4289 | 9.85E-24 | 2.0579 |
| Ldhb | 0.3809 | 8.26E-19 | 2.0235 |
| Acp5 | 0.3837 | 4.45E-19 | 2.0198 |
| Dapk1 | 0.3658 | 2.20E-17 | 1.8704 |
| Itgax | 0.5825 | 2.02E-43 | 1.8680 |
| Csf2rb2 | 0.3459 | 1.11E-16 | 1.7791 |
| H2-DMa | 0.5022 | 2.22E-32 | 1.7642 |
| Trerf1 | 0.3850 | 3.31E-19 | 1.7402 |
| Dmxl2 | 0.4117 | 6.59E-22 | 1.7264 |
| Tgm2 | 0.3663 | 1.99E-17 | 1.6906 |
| Ear2 | 0.5159 | 3.79E-34 | 1.6602 |
| Dok2 | 0.3695 | 1.00E-17 | 1.6373 |
| Flt1 | 0.3655 | 2.34E-17 | 1.5457 |
| Axl | 0.5068 | 5.74E-33 | 1.4698 |
| AU020206 | 0.3723 | 5.40E-18 | 1.4207 |
| Ece1 | 0.3962 | 2.54E-20 | 1.4075 |
| Ear10 | 0.3938 | 4.41E-20 | 1.4069 |
| Il18 | 0.4084 | 1.46E-21 | 1.4021 |
| Clec7a | 0.4959 | 1.37E-31 | 1.3413 |
| Spint1 | 0.3724 | 5.29E-18 | 1.3407 |
| Mpeg1 | 0.5454 | 4.05E-38 | 1.3269 |
| Il1rn | 0.3463 | 1.11E-16 | 1.2922 |
| Clec4n | 0.4454 | 1.45E-25 | 1.2765 |
| Fn1 | 0.4088 | 1.34E-21 | 1.2497 |
| Il1b | 0.4002 | 1.02E-20 | 1.2281 |
| Dst | 0.4484 | 6.72E-26 | 1.1994 |
| F11r | 0.4129 | 4.97E-22 | 1.1982 |
| Nceh1 | 0.3548 | 2.24E-16 | 1.1299 |
| Neurl3 | 0.3721 | 5.74E-18 | 1.1091 |
| Tnfaip2 | 0.3785 | 1.40E-18 | 1.0880 |
| Plet1 | 0.4461 | 1.23E-25 | 1.0845 |

The transient presence of the Mac II cells, and significant overlap with the transcriptomes of Mac I, and III, suggested that Mac II may represent an intermediate population. To determine if cells within these clusters were undergoing gradual transcriptional shifts across time, we performed pseudotime analysis on cells from clusters Mac I, II, and III. Ordering of the cells within the Mac clusters I-III by pseudotime (*Figure 4F*) defined a global trajectory from the Mac I to the Mac III cluster, indicating a correspondence between pseudo- and real time during perinatal development. This global

**Table 3.** Top 35 differentially expressed genes in the Mac III cluster.
ScRNA-Seq was performed on live CD45+ lung cells isolated from E18.5, P1, P7 and P21 B6 pups. Top 35 upregulated genes are shown for Mac III cluster; n = 2 mice.

**Mac III cluster top 35 differentially expressed genes**

| Gene name | Statistic | P value | Log2 fold change |
|---|---|---|---|
| Slc39a2 | 0.6600 | 3.98E-62 | 9.2983 |
| Atp6v0d2 | 0.9533 | 1.14E-151 | 7.1784 |
| Krt19 | 0.7533 | 1.56E-83 | 6.8066 |
| Cd200r4 | 0.7533 | 1.56E-83 | 5.9139 |
| Gpnmb | 0.6967 | 5.32E-70 | 5.0666 |
| Ear1 | 0.6267 | 1.66E-55 | 4.9379 |
| Car4 | 0.7500 | 1.10E-82 | 4.6965 |
| Ly75 | 0.7867 | 2.26E-92 | 4.4964 |
| Bhlhe41 | 0.7800 | 1.51E-90 | 4.4452 |
| Slc7a2 | 0.7933 | 3.16E-94 | 4.1798 |
| Lrp12 | 0.6533 | 9.21E-61 | 3.7640 |
| Ccl6 | 0.8700 | 4.58E-118 | 3.7161 |
| Spp1 | 0.6267 | 1.66E-55 | 3.4907 |
| Serpinb1a | 0.6600 | 3.98E-62 | 3.4778 |
| Mgll | 0.8067 | 4.96E-98 | 3.4554 |
| Il1rn | 0.6567 | 1.93E-61 | 3.2640 |
| Kcnn3 | 0.7600 | 3.03E-85 | 3.2372 |
| Ralgds | 0.7000 | 9.49E-71 | 3.1251 |
| Dst | 0.7633 | 4.13E-86 | 3.1042 |
| Atp10a | 0.6600 | 3.98E-62 | 3.0453 |
| Ccnd2 | 0.7233 | 3.77E-76 | 2.9813 |
| Fabp5 | 0.6233 | 7.18E-55 | 2.9667 |
| Cd36 | 0.7600 | 3.03E-85 | 2.9174 |
| Vat1 | 0.7467 | 7.57E-82 | 2.8682 |
| Myof | 0.7600 | 3.03E-85 | 2.8483 |
| Plet1 | 0.7567 | 2.19E-84 | 2.7934 |
| Hvcn1 | 0.6967 | 5.32E-70 | 2.7689 |
| Fth-ps3 | 0.6933 | 2.94E-69 | 2.7627 |
| Lima1 | 0.6533 | 9.21E-61 | 2.7359 |
| Itgax | 0.7567 | 2.19E-84 | 2.7210 |
| Card11 | 0.6367 | 1.93E-57 | 2.6520 |
| Mir692-2 | 0.7567 | 2.19E-84 | 2.5974 |
| Ctsd | 0.8167 | 5.70E-101 | 2.5402 |
| Klhdc4 | 0.6500 | 4.35E-60 | 2.5367 |
| Mir692-1 | 0.6500 | 4.35E-60 | 2.4553 |

trajectory notwithstanding, there were multiple local attractor states within the Mac II cluster, suggesting that cells gradually shift from Mac I to Mac II, and subsequently remain in that phenotypic state for some time before further committing to a specific fate. We also determined RNA velocity (*La Manno et al., 2018*) and found that the ratio of spliced to unspliced transcripts increased from the Mac I to the Mac II and III clusters along the embedding, with a clear downwards flow suggestive of a continuous development (*Figure 4G*). Of note, in some locations, (e.g. on the left side of Mac I),

**Table 4.** Top 35 differentially expressed genes in the Mac IV cluster.
ScRNA-Seq was performed on live CD45+ lung cells isolated from E18.5, P1, P7 and P21 B6 pups.
Top 35 upregulated genes are shown for the Mac IV cluster; n = 2 mice.

**Mac IV cluster top 35 differentially expressed genes**

| Gene name | Statistic | P value | Log2 fold change |
|---|---|---|---|
| Fcrls | 0.7367 | 2.28E-79 | 12.2139 |
| Pf4 | 0.7900 | 2.70E-93 | 11.4117 |
| Gas6 | 0.6433 | 9.38E-59 | 9.6283 |
| C1qc | 0.8733 | 3.18E-119 | 9.1247 |
| Ccl12 | 0.6400 | 4.28E-58 | 8.7381 |
| C1qa | 0.8700 | 4.58E-118 | 8.3676 |
| Stab1 | 0.8033 | 4.55E-97 | 7.6076 |
| C1qb | 0.8600 | 1.17E-114 | 7.0450 |
| Sdc4 | 0.7200 | 2.32E-75 | 6.3860 |
| Igfbp4 | 0.7433 | 5.15E-81 | 6.2878 |
| Ms4a7 | 0.7200 | 2.32E-75 | 6.0618 |
| Slc40a1 | 0.6033 | 3.72E-51 | 6.0358 |
| Tmem176b | 0.8600 | 1.17E-114 | 5.5402 |
| Maf | 0.7500 | 1.10E-82 | 5.4450 |
| S1pr1 | 0.6367 | 1.93E-57 | 5.3458 |
| Igf1 | 0.6700 | 3.28E-64 | 4.8862 |
| Tmem37 | 0.5900 | 9.02E-49 | 4.8077 |
| Ophn1 | 0.5933 | 2.32E-49 | 4.6672 |
| Tmem176a | 0.7167 | 1.41E-74 | 4.6367 |
| C3ar1 | 0.7100 | 4.98E-73 | 4.5725 |
| Smagp | 0.6700 | 3.28E-64 | 4.5052 |
| Cxcl16 | 0.5800 | 4.98E-47 | 4.2439 |
| Sesn1 | 0.5633 | 3.27E-44 | 4.0183 |
| Apoe | 0.7600 | 3.03E-85 | 4.0149 |
| Hpgds | 0.6333 | 8.62E-57 | 3.9920 |
| Lgmn | 0.7267 | 6.04E-77 | 3.9551 |
| Ms4a4a | 0.5667 | 9.10E-45 | 3.5567 |
| Selenop | 0.6200 | 3.07E-54 | 3.4452 |
| Cx3cr1 | 0.6000 | 1.49E-50 | 3.3845 |
| Abca1 | 0.5433 | 5.69E-41 | 3.2834 |
| Tcf4 | 0.5433 | 5.69E-41 | 3.1404 |
| Blvrb | 0.5533 | 1.42E-42 | 3.1103 |
| Aif1 | 0.6933 | 2.94E-69 | 3.0826 |
| Abca9 | 0.5833 | 1.32E-47 | 3.0325 |
| Pea15a | 0.5400 | 1.91E-40 | 3.0198 |

the arrows seem to exit the clusters without a sink, consistent with metastable local cell states observed in the pseudotime analysis. Taken together, our data suggest that the Mac II cells derive from Mac I and represents a transitional population that may serve a temporal-specific function before later differentiating into Mac III cells and potentially the antigen-presenting subset of Mac IV cells. However, lineage tracing studies employing a *Gal*-promoter driven fluorescent reporter will be required to firmly address the fate of the Mac I cells in the postnatal lung.

**Table 5.** Top 35 differentially expressed genes in the Mac V cluster.
ScRNA-Seq was performed on live CD45+ lung cells isolated from E18.5, P1, P7 and P21 B6 pups.
Top 35 upregulated genes are shown for the Mac V cluster; n = 2 mice.

**Mac V cluster top 35 differentially expressed genes**

| Gene name | Statistic | P value | Log2 fold change |
|---|---|---|---|
| I830127L07Rik | 0.4867 | 1.42E-32 | 7.0634 |
| Ifitm6 | 0.8800 | 1.41E-121 | 6.9362 |
| Pglyrp1 | 0.6867 | 8.67E-68 | 6.2150 |
| Serpinb10 | 0.5333 | 2.09E-39 | 6.1537 |
| Plac8 | 0.9600 | 6.02E-155 | 5.9647 |
| Adgre4 | 0.6033 | 3.72E-51 | 5.8672 |
| Nxpe4 | 0.5000 | 1.89E-34 | 4.8441 |
| Ifitm3 | 0.9267 | 1.17E-139 | 4.6727 |
| Plbd1 | 0.7267 | 6.04E-77 | 3.9525 |
| Susd3 | 0.5033 | 6.29E-35 | 3.9210 |
| Hp | 0.6333 | 8.62E-57 | 3.8550 |
| Pmaip1 | 0.4767 | 3.29E-31 | 3.7016 |
| Adgre5 | 0.7500 | 1.10E-82 | 3.5335 |
| Ifi27l2a | 0.5667 | 9.10E-45 | 3.3416 |
| Atp1a3 | 0.4900 | 4.88E-33 | 3.3206 |
| Ifitm2 | 0.7600 | 3.03E-85 | 3.2359 |
| Xdh | 0.4833 | 4.07E-32 | 3.2003 |
| Pou2f2 | 0.4733 | 9.21E-31 | 3.1216 |
| Rnase6 | 0.5067 | 2.07E-35 | 3.0629 |
| Pla2g7 | 0.6800 | 2.43E-66 | 3.0556 |
| Thbs1 | 0.5767 | 1.86E-46 | 3.0133 |
| Itga4 | 0.6600 | 3.98E-62 | 2.9055 |
| Itgam | 0.5500 | 4.91E-42 | 2.7702 |
| Napsa | 0.6000 | 1.49E-50 | 2.6112 |
| Sorl1 | 0.5033 | 6.29E-35 | 2.5328 |
| Wfdc17 | 0.4967 | 5.64E-34 | 2.5299 |
| Clec4a1 | 0.5300 | 6.81E-39 | 2.4922 |
| Slc11a1 | 0.5267 | 2.20E-38 | 2.4649 |
| Fyb | 0.6400 | 4.28E-58 | 2.4454 |
| Srgn | 0.6633 | 8.14E-63 | 2.2151 |
| Cnn2 | 0.4933 | 1.67E-33 | 2.1880 |
| Samhd1 | 0.6167 | 1.30E-53 | 2.1715 |
| Lst1 | 0.6667 | 1.65E-63 | 2.1340 |
| Cd300a | 0.5100 | 6.78E-36 | 2.1007 |
| Nr4a1 | 0.6100 | 2.24E-52 | 2.0001 |

## Mac V monocytes are characterized by developmental gene expression gradients

The Mac V cluster was characterized by expression of *Plac8* (*Galaviz-Hernandez et al., 2003*; *Figure 1E*), a gene expressed by a recently identified population of 'CD64+ CD16.2+ non-classical monocytes' in the adult mouse lung (*Schyns et al., 2019*), suggesting a monocytic phenotype. Top enriched pathways included cytokine production, phagocytosis, response to virus, and both positive

and negative regulation of the immune response (*Figure 4—figure supplement 1*). Consistent with these results, the Mac V monocytes expressed a panel of unique transcripts induced by type I interferon (IFN), important for modulating host responses to viral pathogens (*Ifitm2, Ifitm3, Ifitm6, Ifi27l2a*) (*Zhao et al., 2018*; *Figure 5A*), as well as genes that constrain inflammation (*Cd300a, Nr4a1, Lst1*) (*Figure 4A* and *Table 5*). This dual immune signature was similar to that seen in Mac II and III, emphasizing the importance of a finely tuned inflammatory response in the developing lung.

Within Mac V we found a gradient of cell states between two distinct phenotypes, with some genes (e.g. *Ccr1* and *Ccr2)* expressed at one end and other genes (e.g. *Ace* and *Slc12a)* expressed at the other end of the spectrum (*Figure 5B*), with a clear correspondence to real developmental time (*Figure 5C*). Pseudotime analysis indicated that cells with an early phenotype upregulated *Ly6C* (*Yona and Jung, 2010*), a gene expressed by classical monocytes, and genes associated with the cytoskeleton (*Actb, Vim*) (*Rubenstein, 1990*; *Capote and Maccioni, 1998*), matrix remodeling (*Fn1, F13a1, Vcan*) (*Wight, 2017*; *Peters et al., 1996*), and reduction-oxidation (*Sod1, Prdx6*) (*Tsang et al., 2018*; *Fisher, 2011*) in keeping with the marked physiological changes and rapid remodeling occurring in the lung during the fetal-neonatal transition (*Figure 5D and E*). Over both pseudo- and real time, this gene expression pattern evolved into an immunomodulatory signature, reflected by the up-regulation of genes associated with macrophage differentiation (*Csf1, Spn*) (*Rojo et al., 2019*), macrophage polarization (*Tgm2, Cd36*) (*Rőszer, 2015*), negative regulation of inflammation (*Ceacam1, Ear2, Lair1*) (*Dankner et al., 2017*), and innate immunity (*Cd300ld, Treml4*) (*Borrego, 2013*). These data indicate that only the late Mac V cells are similar to the CD64+ CD16.2 + non-classical monocytes reported by *Schyns et al., 2019*., while the early Mac V cells represent an additional source of heterogeneity unique to the perinatal lung.

Spatial localization of the early and late populations by detecting either *Ccr2* or *Lair1* in combination with *Ifitm6* and *Cd68* allowed the identification of early Mac V cells within the distal lung parenchyma (*Figure 5F*, *Figure 4—figure supplement 4*). Of note, these *Cd68+ Ifitm6+ Ccr2+* cells were reliably found either along secondary crests or lining the developing alveoli such that one boundary of the cell was always in contact with the airspace. Despite significant changes in the gene expression, *Cd68+ Ifitm6+Lair1+* were similarly found in the distal lung along the alveolar walls (*Figure 5G*). Taken together, these data demonstrate that within the Mac V monocytic phenotype there are functionally distinct, developmentally dynamic cell states that transition during and after birth from tissue remodeling and regulation of immune cell chemotaxis to immunomodulation and pathogen defense.

## Comparison of mac I-V populations with adult murine lung data sets

Next, we determined whether our macrophage populations are similar to recently defined macrophages in the adult lung by directly comparing our data with the macrophage populations annotated by the authors of two previous reports. In comparison to *Schyns et al., 2019*., despite the use of different technologies, we identified clear shared patterns (*Figure 6A–B*). First, our Mac IV cluster embedded close to their CD206+ and CD206- macrophages, and all three express *C1qa*, confirming their identity as interstitial macrophages (IMs). Second, the proportion of IMs expressing *H2-Eb1* is higher in their dataset. Third their monocyte cluster is located closer to our Mac V and both express *Plac8*, indicating a similar cell type. While a few of their cells embedded completely within our Mac V, the majority are in a distinct, yet nearby cluster, possibly reflecting developmental differences. Likewise, although our Mac III and their AMs express Car4 and cluster near one another, they remain in separate clusters, perhaps reflecting either technical or biologic differences. Comparison with Tabula Muris (*Tabula Muris Consortium et al., 2018*) yielded similar results (*Figure 6C–D*), with the Mac IV cells embedding with the cells annotated as macrophages, Mac V embedding with cells annotated as monocytes, and Mac I-III clustering separately. Taken together, these data suggest that Mac IV and the late Mac V are transcriptionally similar to adult lung interstitial macrophages and tissue monocytes, respectively, Mac III are AM with transcriptomic alterations specific to development, and Mac I and II are macrophage populations unique to the perinatal lung.

## Variations in macrophage fc receptor expression

Fc receptors serve as an important link between cellular and humoral immunity by bridging antibody specificity to effector cells (*Hazenbos et al., 1996*) and are therefore a key axis of functional

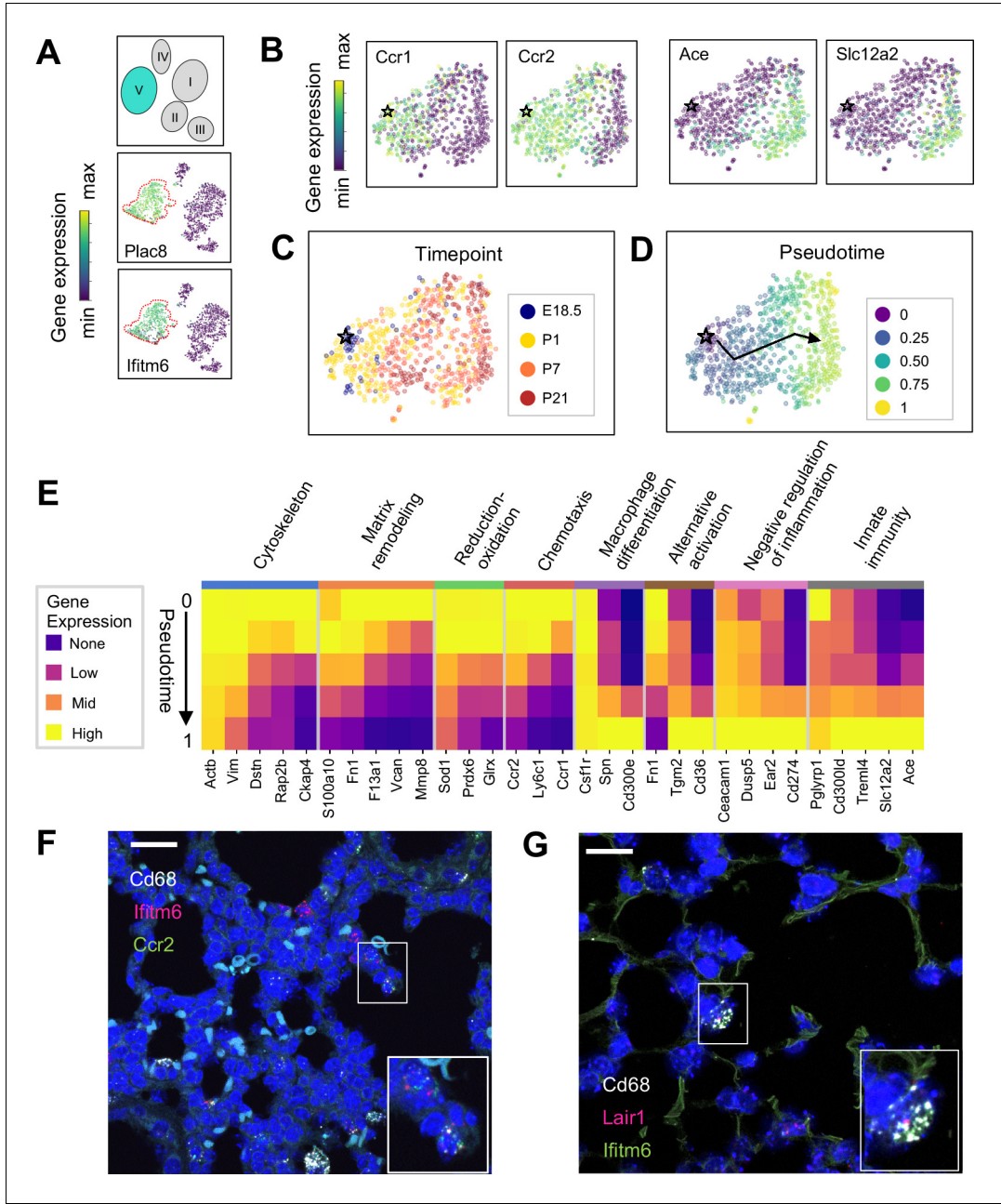

**Figure 5.** The Mac V cluster harbors distinct subpopulations with developmentally regulated gene expression patterns. (**A**) t-SNE plots demonstrating high expression of *Plac8 and Ifitm6* in Mac V cells. (**B**) t-SNE plots of *Ccr1, Ccr2, Ace,* and *Slc12a2* suggesting the presence of two transcriptionally distinct populations within the Mac V cluster. (**C**) t-SNE demonstrating a developmental gradient within the Mac V sub cluster. For all t-SNE embeddings, the color scale is a logarithmic scale with a pseudocount of 0.1 counts per million, normalized to the highest expressing cell. (**D**) Pseudotime analysis with the star indicating the starting point and the arrow denoting the trajectory across pseudotime. (**E**) Heatmap of differentially expressed genes within enriched pathways across pseudotime. The color scale is logarithmic with a pseudocount of 0.1 counts per million, normalized to the highest expressing population for each gene, depicting 0 (none), 33 (low), 67 (mid), and 100% (high) expression compared to the highest expressing population. (**F**) In situ hybridization of *Cd68 (white), Ifitm6 (red),* and *Ccr2 (green),* and to detect the 'early' Mac V subcluster at P1. (**G**) In situ hybridization of *Cd68 (white), Lair1 (red),* and *Ifitm6 (green),* to detect the 'late' Mac V subcluster at P21. Calibration bar = 20 μm for all panels.

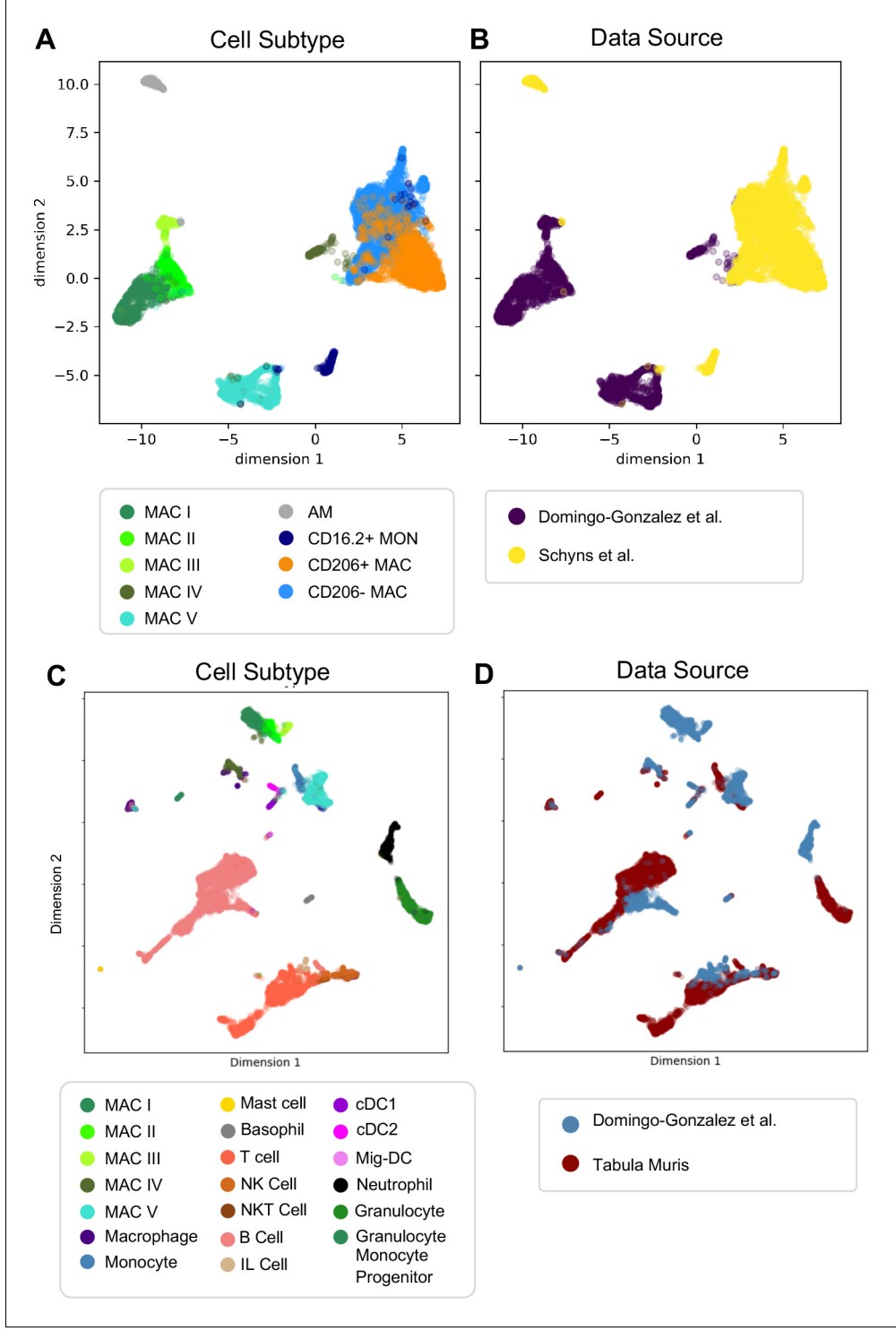

**Figure 6.** Comparison of Macrophage/Monocyte Subtypes in the Perinatal and Adult Lung. Data sets were merged, restricted to genes that were overlapping, normalized to counts per million reads, feature selected and subjected to PCA, followed by UMAP. UMAP plots of (A) cell type and (B) data source using our data combined with Schyns et al. Data sets were merged and processed as described above, and the Northstar algorithm was used to infer the cell subtypes for our cells based on the Tabula Muris Atlas, with UMAP plots of (C) cell type and (D) data source.

heterogeneity within macrophages and monocytes. We evaluated their expression across Mac I–V and found specific patterns (*Figure 7*). Both *Fcgr3* (encoding FcγRIII) and *Fcer1g* (FcεRIγ) were widely expressed by all five clusters, while *Fcer1a* and *Fcer2a* were not expressed by any cluster. Expression of *Fcgr1*, a high affinity receptor for IgG important for the endocytosis of soluble IgG, phagocytosis of immune complexes, and delivery of immune complexes to APC (*Barnes et al., 2002*) was highly expressed by Mac I and Mac IV, and in the early subcluster of Mac V. In contrast, *Fcgr4*, encoding an Fc receptor able to bind IgE that promotes allergic lung inflammation was highly expressed by the late sub-cluster of Mac V, in agreement with data from adult mice (*Schyns et al., 2019*; *Mancardi et al., 2008*; *Hirano et al., 2007*). *Fcgrt*, encoding the neonatal Fc receptor (FcRn), an Fc receptor with a key role in IgG recycling, was highest in Mac IV cells.

## Dendritic cell subtypes and granulocytes are primed for rapid pathogen response

Dendritic cells (DCs) play multiple roles in the immune system including antigen presentation and regulation of tolerance and can be distinguished from other mononuclear phagocytes by the expression of *Zbtb46* (*Meredith et al., 2012*) and Flt3 (*Waskow et al., 2008*). We found three clusters of cells (*Figure 8A* and *Tables 6–8*) expressing *Zbtb46* and *Flt3* (*Figure 8—figure supplement 1*). Classical DC1 (cDC1) cells expressed *Itgae* or CD103 (*Figure 8B*), which promotes antiviral immunity and may confer the ability for antigen cross-presentation (*Helft et al., 2012*), and a number of additional genes characteristic of cDC1 including *Irf8*, *Xcr1*, and *Cadm1* (*Figure 8—figure supplement 1*). The next DC cluster expressed many genes described to be specific for classical DC2 cells, including *Itgam*, *Sirpa*, and *Irf4* (*Figure 8—figure supplement 1*). Although cDC2 expressed *Cd209a* or DC-SIGN, a gene found in monocyte-derived inflammatory DC exposed to lipopolysaccharide (*Cheong et al., 2010*), and low levels of *Fcer1g*, other genes expressed by monocyte-derived DC, including *Fcgr1*, *Ly6c1*, and *Ly6c2* were practically absent (*Figure 8—figure supplement 1*). Given that cDC2 cells can also express *Cd209a*, the identity of these cells was most consistent with cDC2. The final DC cluster expressed melanoregulin (*Mreg*), a modulator of lysosomal hydrolase maturation (*Damek-Poprawa et al., 2009*; *Figure 8B*). *Mreg* had not been previously identified as a marker for DC subsets, hence we examined other genes expressed by this cluster and identified *Cacnb3*, a voltage dependent $Ca^{2+}$ channel found in stimulated Langerhans cells (*Bros et al., 2011*); *Fscn1*, which contributes to dendrite formation in maturing DC (*Al-Alwan et al., 2001*); *Ccl5*, an important chemoattractant for DC and T cells; and *Ccr7*, a chemokine receptor associated with trafficking to the draining lymph node (*Förster et al., 2008*), therefore we annotated this cluster as migratory (mig-DC) (*Figure 8—figure supplement 1* and *Tables 6–8*). Quantification of the relative abundances identified cDC1 as the predominant population in the embryonic lung. cDC1 persisted postnatally to comprise between 1–2% of total lung immune cells (*Figure 7C*). cDC2 was present at low frequency during the first week of life and increased in abundance by P21. The

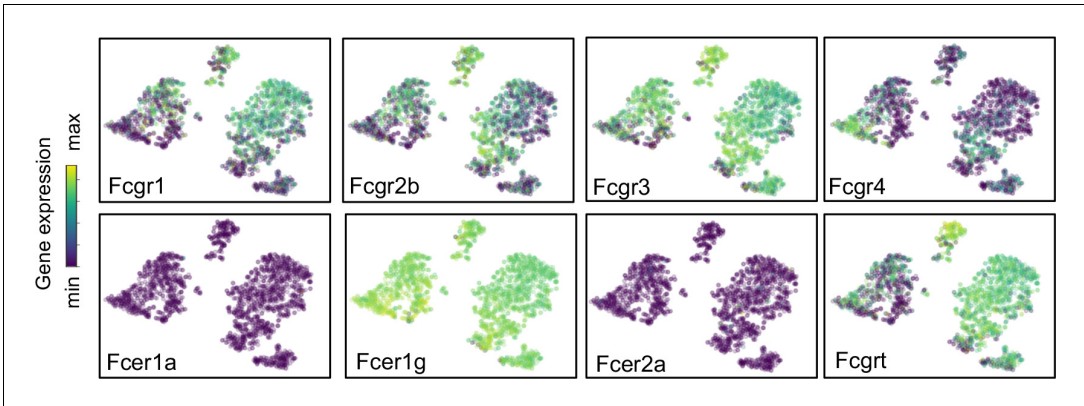

**Figure 7.** Expression of Fc receptors across macrophage clusters. t-SNE plots of the expression of Fc receptors *Fcgr1, Fcgr2b, Fcgr3, Fcgr4, Fcer1a, Fcer1g, Fcer2a, Fcgrt*. For all tSNE embeddings, the color scale is a logarithmic scale with a pseudocount of 0.1 counts per million, normalized to the highest expressing cell.

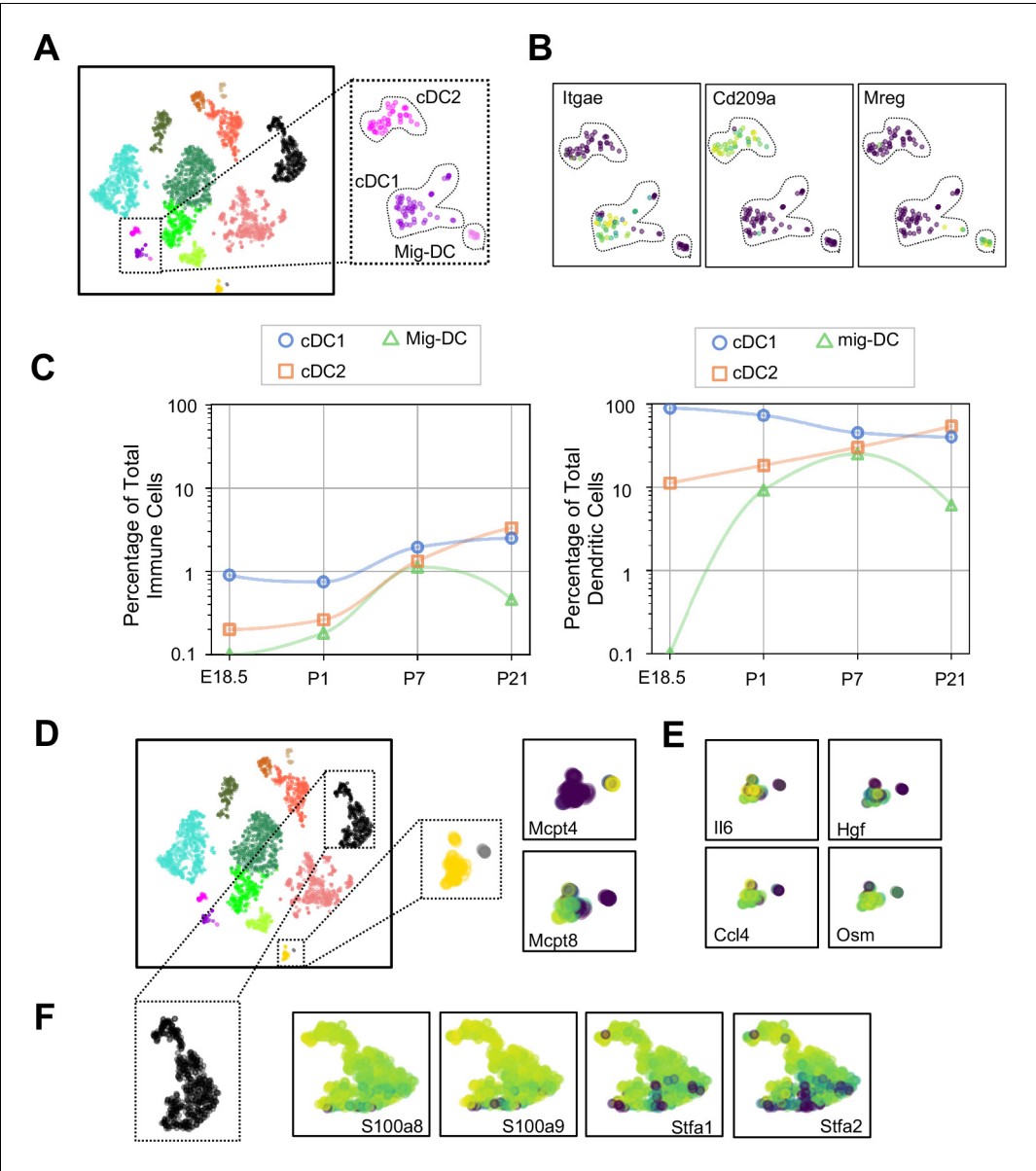

**Figure 8.** Multiple dendritic cell populations and lung granulocytes are primed for rapid pathogen response. (**A**) Colored schematic and lung immune cell clustering demonstrating three separate clusters of DCs. (**B**) t-SNE plots of genes discriminating the three DC subclusters including *Itgae* (cDC1), *Cd209a* (cDC2) *and Mreg* (mig-DC). (**C**) Quantification of specific DC subpopulations relative to total immune cells (left) or total DC (right). (**D**) Colored schematic and lung immune cell clustering demonstrating the basophil, mast cell and neutrophil clusters, with high magnification of basophil and mast cell clusters and t-SNE plots of *Mcpt4* and *Mcpt8*. (**E**) t-SNE plots of *Il6*, *Hgf*, *Ccl4*, and *Osm* in the basophil cluster. (**F**) High magnification of the neutrophil cluster with t-SNE plots of neutrophil specific genes *S100a8*, *S100a9*, *Stfa1*, and *Stfa2*. For all t-SNE embeddings, the color scale is a logarithmic scale with a pseudocount of 0.1 counts per million, normalized to the highest expressing cell. The online version of this article includes the following figure supplement(s) for figure 8:

**Figure supplement 1.** Expression of dendritic cell associated genes.

mig-DC were undetectable before birth and became more abundant postnatally. Taken together, these data suggest cDC1and cDC2 are classical DC, and the mig-DC cluster a form of migratory mature DCs.

Mast cells and basophils are similar in development and function and serve as fast responders to specific immune challenges (*Voehringer, 2013*). Two immune clusters highly expressed *Cpa3* that

**Table 6.** Top 20 differentially expressed genes in the Mig-DC vs cDC2 cluster.
ScRNA-Seq was performed on live CD45+ lung cells isolated from E18.5, P1, P7 and P21 B6 pups.
Top 20 differentially expressed genes are shown for the migDC vs cDC2 cluster; n = 2 mice.

**Mig-DC versus cDC2 cluster top 20 differentially expressed genes**

| Gene name | Statistic | P. value | Log2 fol d change |
|-----------|-----------|----------|-------------------|
| Ccl5 | 1.0000 | 3.45E-13 | 16.9849 |
| Il12b | 0.7857 | 6.97E-07 | 15.3906 |
| Fscn1 | 1.0000 | 3.45E-13 | 14.5028 |
| Nudt17 | 0.9286 | 2.48E-10 | 13.7990 |
| Zmynd15 | 0.9286 | 2.48E-10 | 13.5523 |
| Il15ra | 0.9286 | 2.48E-10 | 13.5174 |
| Cacnb3 | 1.0000 | 3.45E-13 | 13.4891 |
| Cd63-ps | 1.0000 | 3.45E-13 | 13.4587 |
| H2-M2 | 0.8571 | 2.06E-08 | 13.4381 |
| Tnfrsf4 | 0.7143 | 1.31E-05 | 13.4073 |
| Serpinb6b | 0.9286 | 2.48E-10 | 13.2991 |
| AW112010 | 0.8571 | 2.06E-08 | 13.1876 |
| Laptm4b | 0.8571 | 2.06E-08 | 12.9752 |
| Lad1 | 0.7143 | 1.31E-05 | 12.8948 |
| Cd63 | 1.0000 | 3.45E-13 | 12.8680 |
| Dnase1l3 | 0.7143 | 1.31E-05 | 12.7389 |
| Apol7c | 0.5000 | 7.75E-03 | 12.7302 |
| Tm4sf5 | 0.5000 | 7.75E-03 | 12.6086 |
| Spint2 | 0.9286 | 2.48E-10 | 12.2644 |
| Zfp872 | 0.6429 | 1.57E-04 | 11.4802 |

could be further distinguished as mast cells and basophils by the expression of *Mcpt4* and *Mcpt8*, respectively (*Lilla et al., 2011*; *Immunological Genome Project Consortium et al., 2016*; *Figure 8D*). Mast cells expressed *Tpsb2*, which is secreted upon bacterial challenge (*Thakurdas et al., 2007*), and the peptidases chymase (*Cma1*) and tryptase (*Tspab1*) (*Table 9*). Lung resident basophils express a unique signature distinct from circulating basophils and play a key role in promoting AM differentiation (*Cohen et al., 2018*). Our basophils generally shared expression of many genes with lung resident basophils, including *Il6*, *Hgf*, *Ccl4* and *Osm* (*Figure 8E*, *Table 10*). We also identified a neutrophil cluster distinguished by expression of *S100a8* and *S100a9*, which are released during inflammation (*Wang et al., 2018*), and *Stfa1* and *Stfa2*, cysteine proteinase inhibitors important for antigen presentation (*Hsieh et al., 2002*; *Figure 8F* and *Table 11*). Our data agree with prior work demonstrating that lung resident basophils express signaling molecules important for interaction with neighboring cells (*Cohen et al., 2018*), and that mast cells and neutrophils are primed for rapid innate immune responses upon pathogen challenge.

## Naïve lymphocytes populate the lung at birth

Lymphocytes including ILC2s (expressing *Areg*), NK cells (*Gzma*), B cells (*Ms4a1*) and T cells (*Cde3*) were present in the lung at low frequencies (approximately 2% of immune cells) prior to birth and increased in frequency after birth. By P21, lymphocytes comprised 60% of total immune cells (*Figure 1F and G*), with B cells representing 30% and T cells 15% of the total (*Figure 9A*).

B cells in the embryonic lung were rare and the majority expressed proliferation markers. After birth, B cell abundance increased, but the proliferating fraction decreased (*Figure 9B*). The cumulative distribution of V and J loci germline similarity revealed no somatic hypermutation (*Figure 9C*) and most B cells expressed *Ighm* and *Ighd* indicative of an IgM isotype (*Figure 9D*). Few B cells expressed activation markers (e.g. *Aicda*, *Tbx21*, *Prdm1*, and *Ebi3*, *Figure 9—figure supplement*

**Table 7.** Top 20 differentially expressed genes in the Mig-DC vs cDC1 cluster.
ScRNA-Seq was performed on live CD45+ lung cells isolated from E18.5, P1, P7 and P21 B6 pups. Top 20 differentially expressed genes are shown for the mig-DC vs cDC1 cluster; n = 2 mice.

**Mig-DC versus cDC1 cluster top 20 differentially expressed genes**

| Gene name | Statistic | P. value | Log2 fold change |
|---|---|---|---|
| H2-M2 | 0.8571 | 2.34E-09 | 14.0574 |
| Tnfrsf4 | 0.7143 | 3.71E-06 | 13.6282 |
| Zfp872 | 0.6429 | 5.96E-05 | 11.348 |
| Arc | 0.7143 | 3.71E-06 | 10.9369 |
| Gbp8 | 0.6429 | 5.96E-05 | 10.6692 |
| Ramp3 | 0.5 | 4.51E-03 | 10.6263 |
| Insl6 | 0.6429 | 5.96E-05 | 10.5787 |
| Gbp4 | 0.9286 | 1.19E-11 | 10.4833 |
| Adcy6 | 0.8201 | 2.13E-08 | 10.4176 |
| Atxn1 | 0.7143 | 3.71E-06 | 10.3352 |
| Ankrd33b | 0.9286 | 1.19E-11 | 10.1242 |
| Dscaml1 | 0.6429 | 5.96E-05 | 9.8945 |
| Src | 0.5 | 4.51E-03 | 9.7618 |
| Tmem150c | 0.6429 | 5.96E-05 | 9.566 |
| Slc4a8 | 0.7116 | 4.01E-06 | 9.5518 |
| Nudt17 | 0.9101 | 6.10E-11 | 9.5137 |
| Il21r | 0.6958 | 7.81E-06 | 9.3355 |
| Fas | 0.6243 | 1.12E-04 | 9.3023 |
| Clec2i | 0.5 | 4.51E-03 | 9.2653 |
| Cacnb3 | 0.9815 | 2.44E-13 | 8.6751 |

*1*; *Wang et al., 2012*; *Pone et al., 2012*; *Ma et al., 2019*; *Cattoretti et al., 2005*), suggesting that most postnatal B cells remain naïve through late alveolarization. To test the clonality of the B cell repertoire at birth, we assembled the heavy and light chain loci and performed t-SNE on a feature-selected transcriptome limited to over-dispersed genes in B cells (*Figure 9—figure supplement 1*). Proliferating cells clustered together but candidate clonal families did not, suggesting primarily homeostatic B cell proliferation rather than clonal expansion (*van Zelm et al., 2007*).

The majority of T cells expressed *Trac*, suggesting αβ identity, with a few *Trac-* cells expressed *Tcrg-C4*, suggesting γδ T cell identity (*Figure 9—figure supplement 1*). T cell receptor diversity showed no sign of clonal expansion (data not shown). Outside the thymus, αβ T cells are usually either CD4+ (with subgroups Tbet+ or T helper (Th) 1, Gata3+ or Th2, and FoxP3+ or Th17) or CD8 +, however we characterized T cell heterogeneity in neonatal lungs by flow cytometry and found that 85–90% of CD3+ cells were CD4- CD8- at both P1 and P21 (*Figure 9E*), confirming an earlier report suggesting this is a neonatal-specific phenotype. mRNA expression analysis qualitatively confirmed this finding (*Figure 9—figure supplement 1*). The frequency of total T cells was similar at P1 and P21 (*Figure 9—figure supplement 1*), however several T cell subsets (CD8[+], CD4+ Th1, and Treg cells) increased by P21 while other subsets remained constant (*Figure 9F and G*). In response to stimulation with phorbol myristate acetate (PMA) and ionomycin, a greater number of CD4[+] Th2 cells at P21 produced IL-4 as compared to P1 (28.3 ± 19.7 vs. 8.6 ± 5.7, p=0.0047) (*Figure 8H*). Few CD4+ Th1, Treg, and Th17 cells produced IFNγ, IL-10, IL-17 upon stimulation, at both P1 and P21 (*Figure 9H*, *Figure 9—figure supplement 1*).

**Table 8.** Top 20 differentially expressed genes in the cDC2 vs cDC1 cluster.
ScRNA-Seq was performed on live CD45+ lung cells isolated from E18.5, P1, P7 and P21 B6 pups.
Top 20 differentially expressed genes are shown for the cDC2 vs cDC1 cluster; n = 2 mice.

**cDC2 versus cDC1 cluster top 20 differentially expressed genes**

| Gene name | Statistic | P. value | Log2 fold change |
|---|---|---|---|
| *Mgl2* | 0.7857 | 5.55E-15 | 12.5236 |
| *Cd209d* | 0.5476 | 4.76E-07 | 12.1889 |
| *Cd209b* | 0.3333 | 7.77E-03 | 12.1445 |
| *Ms4a6d* | 0.5714 | 1.14E-07 | 11.7772 |
| *Ms4a4c* | 0.5476 | 4.76E-07 | 11.2462 |
| *Plaur* | 0.3810 | 1.42E-03 | 11.0707 |
| *Casp4* | 0.3810 | 1.42E-03 | 10.0788 |
| *Il1rl1* | 0.6190 | 5.17E-09 | 10.0357 |
| *Cd300lg* | 0.3333 | 7.77E-03 | 9.9463 |
| *Cd33* | 0.4762 | 2.21E-05 | 9.7976 |
| *Emilin2* | 0.4048 | 5.52E-04 | 9.5336 |
| *Il1rl2* | 0.3810 | 1.42E-03 | 9.5332 |
| *Creb5* | 0.3915 | 9.39E-04 | 9.2747 |
| *Cd209c* | 0.4762 | 2.21E-05 | 9.0161 |
| *Cd209a* | 0.9577 | 3.33E-16 | 9.0160 |
| *Ccl9* | 0.6772 | 7.43E-11 | 9.0010 |
| *Fcrls* | 0.3810 | 1.42E-03 | 8.9226 |
| *Il18rap* | 0.3915 | 9.39E-04 | 8.7118 |
| *Wfdc17* | 0.7249 | 1.46E-12 | 8.5219 |
| *Tent5a* | 0.6534 | 4.52E-10 | 8.2999 |

## Discussion

At birth, the lung undergoes marked physiological changes as it transitions from a fluid-filled, hypoxic environment to an air-filled, oxygen-rich environment. How these changes affect immune populations during this transition and the ensuing period of rapid postnatal lung growth remains unclear. Our study demonstrates a rapid increase in immune cell heterogeneity, especially within macrophages and monocytes. We identified five macrophage/monocyte subpopulations, each expressing a specific gene signature, spatial localization, and putative functions. Mac I, the predominant immune cell present just before birth, were highly proliferative, and completely encircled small blood vessels, suggesting a previously unrecognized role for lung macrophages in modulating lung vascular growth or remodeling during development. During the first week of life, a transitory population (Mac II) emerged from Mac I and later disappeared, transitioning into either an alveolar (Mac III) or Mac IV macrophage phenotype. One interstitial macrophage population (Mac IV) expressed complement proteins and other antibacterial molecules. Another interstitial population (Mac V) expressed antiviral molecules and spanned a gradient between two extreme phenotypes, one that expressed high levels of homeostatic genes during early postnatal development, and a second with immunomodulatory functions that resembles previously reported nonclassical monocytes. Lymphocytes increased in abundance from almost zero before birth to more than half of lung immune cells by P21, but maintained a naive phenotype skewed toward type II immunity and with predominantly Cd4- Cd8- T cells.

This comprehensive study has far-reaching implications for lung biology. Resident tissue macrophage populations are established during development, wherein progenitors undergo differentiation guided by the tissue-specific microenvironment (*Sommerfeld et al., 2019*). However, definitive data regarding the full complexity of lung resident macrophages and monocytes, their specific roles and

**Table 9.** Top 25 differentially expressed genes in the mast cell cluster.
ScRNA-Seq was performed on live CD45+ lung cells isolated from E18.5, P1, P7 and P21 B6 pups.
Top 25 upregulated genes are shown for the mast cell cluster; n = 2 mice.

**Mast cell top 25 differentially expressed genes**

| Gene name | Statistic | P. value | Log2 fold change |
|---|---|---|---|
| Tpsb2 | 1.0000 | 8.9E-16 | 18.7534 |
| Mcpt4 | 0.8750 | 2.4E-07 | 17.4253 |
| Tpsab1 | 0.8750 | 2.4E-07 | 15.5899 |
| Cma1 | 1.0000 | 8.9E-16 | 15.0710 |
| Slc45a3 | 0.7500 | 5.2E-05 | 11.7029 |
| Rprm | 0.8750 | 2.4E-07 | 11.6969 |
| Hs6st2 | 0.7500 | 5.2E-05 | 11.1665 |
| Maob | 0.8683 | 3.4E-07 | 10.7834 |
| Cpa3 | 0.9900 | 1.8E-13 | 10.6394 |
| Ednra | 0.7500 | 5.2E-05 | 10.2131 |
| Gata1 | 0.7500 | 5.2E-05 | 10.0639 |
| Kit | 1.0000 | 8.9E-16 | 9.2771 |
| Mlph | 0.7467 | 5.9E-05 | 8.5668 |
| Cma2 | 0.7500 | 5.2E-05 | 8.5455 |
| Cyp11a1 | 0.9933 | 4.9E-14 | 8.4308 |
| Tph1 | 0.8717 | 2.8E-07 | 8.3091 |
| Rab27b | 0.8717 | 2.8E-07 | 8.1436 |
| Il1rl1 | 0.9733 | 1.4E-11 | 7.7399 |
| Itga2b | 0.8350 | 1.8E-06 | 7.6658 |
| Gata2 | 0.9933 | 4.9E-14 | 7.5597 |
| Poln | 0.7367 | 8.2E-05 | 7.2024 |
| Slc18a2 | 0.9600 | 1.4E-10 | 6.8570 |
| Stard13 | 0.7467 | 5.9E-05 | 6.7772 |
| Adora3 | 0.7033 | 2.3E-04 | 6.4180 |
| Kcne3 | 0.8050 | 6.6E-06 | 6.4033 |

functions, and how they change across development remain elusive. Although the advent of single cell transcriptomics has provided increased resolution to detect previously unrecognized immune cell populations, consensus regarding the diversity and function of lung resident macrophages has not been achieved. *Cohen et al., 2018* recently performed single cell transcriptomics of the developing mouse lung from E12.5 until P7, and identified a total of three macrophage populations, and one population of resident monocytes, with alveolar macrophages representing the sole macrophage population present in the lung after P7. In contrast, Schyns et al. identified two distinct interstitial macrophages in the adult lung, and a population of nonclassical monocytes in addition to alveolar macrophages (*Schyns et al., 2019*). Although our results are more consistent with the report of Schyns et al, there are a number of key differences. First, the total heterogeneity in the perinatal lung far exceeds the adult lung, with the presence of two unique macrophage clusters (Mac I and Mac II, and alveolar macrophage (Mac III) and monocyte derived clusters (Mac V) that are transcriptionally distinct from their adult counterparts. Second, both the Mac IV and Mac V cluster harbor significant internal heterogeneity (in the case of Mac V, corresponding to developmental time) that cannot be easily split into transcriptionally distinct 'subclusters'. The cells within Mac IV appear similar to the CD206+ and CD206- macrophage populations reported by Schyns et al. Although in the Schyns report those populations were reported to be distinct clusters, there was significant overlap in gene expression between the two, more consistent with our data suggesting

**Table 10.** Top 25 differentially expressed genes in the basophil cluster.

ScRNA-Seq was performed on live CD45+ lung cells isolated from E18.5, P1, P7 and P21 B6 pups. Top 25 upregulated genes are shown for the basophil cluster; n = 2 mice.

**Basophil top 25 differentially expressed genes**

| Gene name | Statistic | P. value | Log2 fold change |
| --- | --- | --- | --- |
| *Mcpt8* | 0.7347 | 2.22E-16 | 15.6663 |
| *Ms4a2* | 0.8676 | 2.22E-16 | 12.3460 |
| *Gata2* | 0.9084 | 2.22E-16 | 9.7841 |
| *Il4* | 0.6872 | 2.22E-16 | 8.7980 |
| *Cd200r3* | 0.9525 | 2.22E-16 | 8.6767 |
| *Il6* | 0.8309 | 2.22E-16 | 8.5795 |
| *Ccl4* | 0.9021 | 2.22E-16 | 7.0748 |
| *Ifitm1* | 0.9363 | 2.22E-16 | 6.8128 |
| *Cyp11a1* | 0.9729 | 2.22E-16 | 6.6271 |
| *Ccl3* | 0.9225 | 2.22E-16 | 5.9632 |
| *Hgf* | 0.7326 | 2.22E-16 | 5.7653 |
| *Csf1* | 0.8921 | 2.22E-16 | 5.4838 |
| *Ccl9* | 0.8825 | 2.22E-16 | 5.3108 |
| *Ifitm7* | 0.7559 | 2.22E-16 | 5.1284 |
| *Aqp9* | 0.7563 | 2.22E-16 | 5.0857 |
| *Rab44* | 0.6534 | 2.22E-16 | 5.0335 |
| *Hdc* | 0.9133 | 2.22E-16 | 4.9976 |
| *Cd69* | 0.7454 | 2.22E-16 | 4.9828 |
| *Cdh1* | 0.7197 | 2.22E-16 | 4.9643 |
| *Il18r1* | 0.6539 | 2.22E-16 | 4.5920 |
| *Csf2rb2* | 0.6930 | 2.22E-16 | 4.3472 |
| *Lilr4b* | 0.8788 | 2.22E-16 | 4.1383 |
| *Rgs1* | 0.7001 | 2.22E-16 | 4.0712 |
| *Il18rap* | 0.7571 | 2.22E-16 | 4.0030 |
| *Osm* | 0.6559 | 2.22E-16 | 3.6933 |

these are not separate populations but rather a phenotypic continuum. In situ validation confirmed the presence of all five subpopulations and localized each to defined locations in the lung including the alveolar lumen, around vessels and airways, or within the distal lung parenchyma. A greater understanding of macrophage and monocyte function at birth provides an essential framework for interpreting how lung injury and developmental defects alter specific immune subpopulations and eventually influence lung growth and development.

Another key finding in our study was the unexpected presence of embryonic lung macrophages encircling small blood vessels. Vascular growth is a key driver of distal lung growth during the late saccular and alveolar stages of development (*Thébaud and Abman, 2007*). Macrophages support angiogenesis in other organs, promoting blood vessel formation or expansion, providing survival and migratory cues to endothelial cells, and facilitating bridging of vascular sprouts (*Baer et al., 2013*). In the developing hindbrain, macrophages are in close contact with endothelial cells, serving to promote vascular anastomosis (*Fantin et al., 2010*). Similarly, in the developing retinal vasculature, microglia connect adjacent endothelial tips cells to increase vascular plexus complexity (*Rymo et al., 2011*). These embryonic bridging macrophages secrete numerous genes shared by the perivascular macrophages that drive tumor angiogenesis including the angiopoietin receptor, *Tek*, the VEGF co-receptor *Nrp1*, growth factors (*Fgf2*, *Pgf*) and MMPs (*Mmp2*, *Mmp9*). Although the perivascular macrophages we observed in the embryonic lung expressed low levels of *Nrp1*, they

**Table 11.** Top 25 differentially expressed genes in the neutrophil cluster.
ScRNA-Seq was performed on live CD45+ lung cells isolated from E18.5, P1, P7 and P21 B6 pups. Top 25 upregulated genes are shown for the neutrophil cluster; n = 2 mice.

**Neutrophil top 25 differentially expressed genes**

| Gene name | Statistic | P. value | Log2 fold change |
|---|---|---|---|
| *Retnlg* | 0.7833 | 1.9E-91 | 16.7841 |
| *Stfa2* | 0.9067 | 1.6E-131 | 13.3347 |
| *BC100530* | 0.9033 | 3.1E-130 | 11.5407 |
| *Mmp9* | 0.8867 | 5.5E-124 | 11.2071 |
| *Asprv1* | 0.7500 | 1.1E-82 | 10.6669 |
| *Stfa2l1* | 0.9300 | 4.5E-141 | 8.5427 |
| *S100a9* | 0.9567 | 2.7E-153 | 8.3546 |
| *2010005H15Rik* | 0.8467 | 2.9E-110 | 8.1305 |
| *BC117090* | 0.8800 | 1.4E-121 | 7.8635 |
| *Slpi* | 0.8767 | 2.1E-120 | 7.5069 |
| *Stfa1* | 0.8967 | 1.1E-127 | 6.8526 |
| *Il1r2* | 0.7133 | 8.4E-74 | 6.6465 |
| *S100a8* | 0.9467 | 1.6E-148 | 6.2103 |
| *Slc7a11* | 0.6933 | 2.9E-69 | 5.9640 |
| *Csf3r* | 0.7400 | 3.5E-80 | 5.7411 |
| *Cxcr2* | 0.7433 | 5.2E-81 | 5.6599 |
| *Wfdc21* | 0.7733 | 9.5E-89 | 5.3370 |
| *Pglyrp1* | 0.8367 | 4.4E-107 | 4.9416 |
| *Mxd1* | 0.6800 | 2.4E-66 | 4.5560 |
| *Hdc* | 0.8267 | 5.5E-104 | 4.2867 |
| *Grina* | 0.7033 | 1.7E-71 | 4.0795 |
| *Il1b* | 0.6567 | 1.9E-61 | 3.8434 |
| *Thbs1* | 0.6733 | 6.5E-65 | 2.7913 |
| *S100a11* | 0.7600 | 3.0E-85 | 2.6263 |
| *Srgn* | 0.6600 | 4.0E-62 | 2.3914 |

appear distinct from the macrophages that influence retinal and hindbrain angiogenesis, expressing a unique set of ECM remodeling and angiogenic genes, including genes that may modulate vascular tone and permeability. Whether the macrophages form non-contiguous rings around the vessel, or sheets that completely encircle the full length of the vessel is not clear and would require antibody-based immunofluorescence with deep imaging and reconstruction of thick tissue sections. The distinctive location of these macrophages and their gene signature imply a role in vascular development. Furthermore, these encircling macrophages disappeared after birth, suggesting a function temporally restricted to prenatal development. Future studies to selectively target this subpopulation will be required to further establish their function and to delineate the signals responsible for the cessation of the macrophage-vascular interaction after birth.

Our data also revealed three distinct dendritic cell populations with transcriptional signatures indicative of classical and migratory dendritic cell phenotypes, respectively. Moreover, we identified distinct (*Itgae* and *Cd209a*) and novel (*Mreg*) markers superior to classical markers (*Zbtb46* and *Flt3*) to distinguish dendritic subpopulations in the postnatal lung. Functionally, both *Itgae* and *Cd209a/* DC-SIGN can induce T cell immunity (*Soilleux et al., 2002*; *Merad et al., 2013*), suggesting that lung DCs immediately post-birth should be able to cause an effective adaptive immune response. However, despite the apparent signaling readiness of antigen-presenting DCs, T and B cell compartments showed an overall naive and rarely proliferative phenotype, lacking any clonal structure and

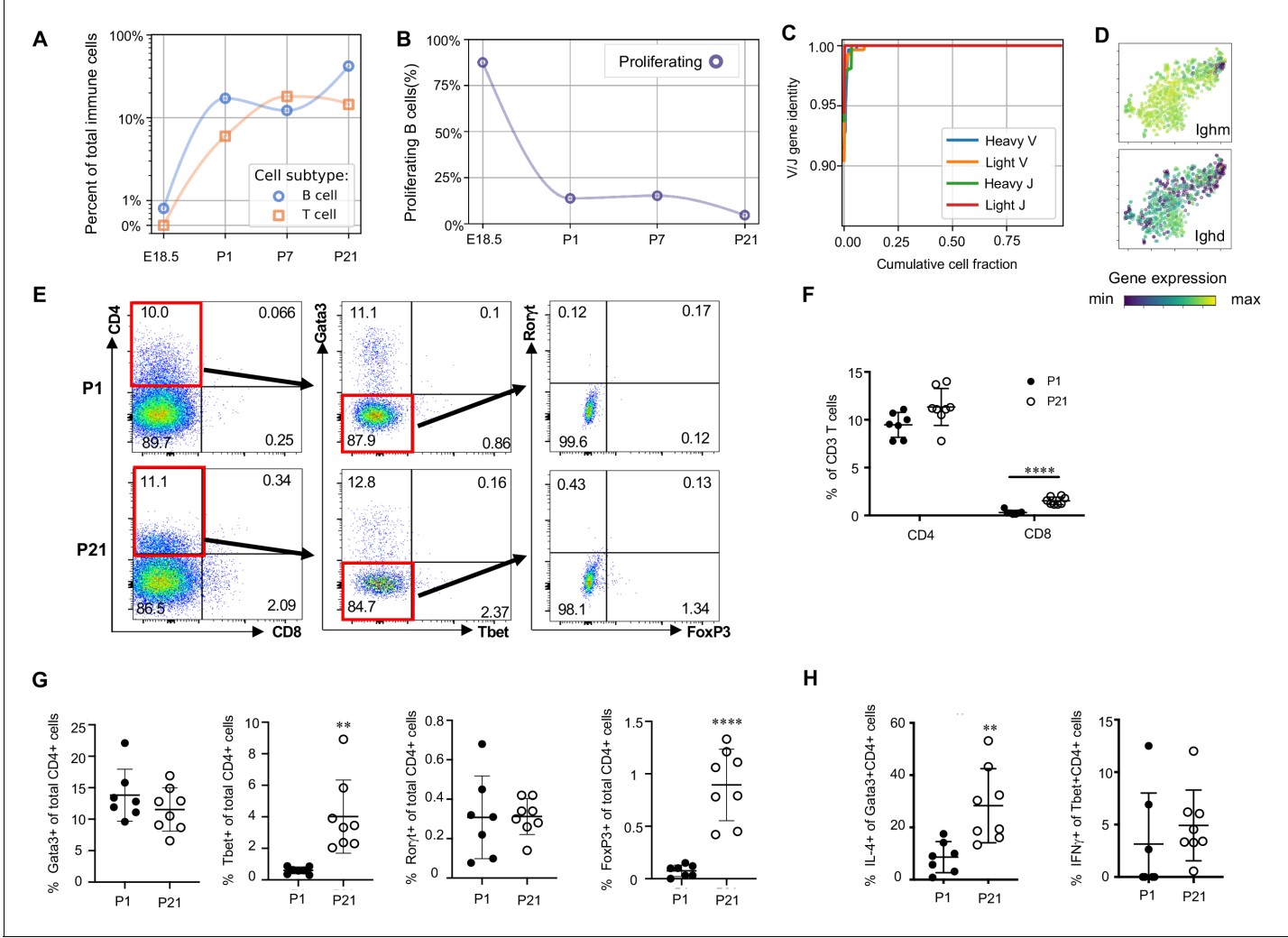

**Figure 9.** Lymphocytes populate the lung at birth but remain naïve during the first three weeks of life (A) Quantification of the abundance of B and T cells at each developmental timepoint expressed on a log (*Rodríguez-Castillo et al., 2018*) scale as percentage of total immune cells. (B) Quantification of the percentage of proliferating B cells at each developmental timepoint. (C) B cell heavy and light variable (V) and joining (J) gene identity and their cumulative cell fraction. (D) t-SNE plot of *Ighm* and *Ighd*, with the logarithmic color scale with a pseudocount of 0.1 counts per million, normalized to the highest expressing cell. (E–H) At P1 (n = 7 mice) and P21 (n = 8 mice), lungs were processed to a single-cell suspension, and flow cytometry was used to assess frequencies of (F) CD4+CD3+ and CD8+CD3+ T cells, (G) Gata3+, Tbet+, Rorγt+ and Foxp3+ CD4+ T cells, and (H) IL-4-producing Gata3+CD4+ T cells or IFNγ-producing Tbet+CD4+ T cells. Data shown as mean ± SD, **p<0.01, ****p<0.00001 by Student's *t* test. The online version of this article includes the following figure supplement(s) for figure 9:

**Figure supplement 1.** Transcriptional and flow cytometric profiling of lymphocytes.

with most αβ T cells double negative (DN) at the protein level for both CD4 and CD8, key signaling components for cell-mediated immunity. Though consistent with prior evidence showing a high proportion of pulmonary lymphocytes with unconventional phenotypes (*Augustin et al., 1989*), the observation of widespread DN T cells points to a yet undetermined function, perhaps related to specific immunoregulatory functions during infectious disease (*Zhang et al., 2000*).

The predominant stimuli that drive the increase in lung immune diversity after birth are not clear. Multiple, marked physiologic changes occur at birth including an abrupt change from hypoxia to normoxia, establishment of an air-liquid interface, cyclic stretch of the lung parenchyma, and increased shear stress in the pulmonary vasculature from the dramatic increase in pulmonary blood flow. These alterations in force and oxygen tension may influence the precise and coordinated process of lung immune cell differentiation or phenotype. For example, continuous compressive force is

an essential driver of osteoclast differentiation (*Sanuki et al., 2010*). Cyclic stretch modulates alveolar macrophage phenotype in vitro, potentially increasing (*Pugin et al., 1998*) or suppressing inflammatory cytokine production (*Maruyama et al., 2019*). Reactive oxygen species influence the expansion, maturation, and antigen uptake and processing in DCs (*Sheng et al., 2010*), while hypoxia promotes macrophage differentiation (*Grodzki et al., 2013*) and influences macrophage polarization, resulting in alterations in inflammation, metabolism and angiogenesis (*Escribese et al., 2012*). However, how an abrupt transition from hypoxia to normoxia affects the phenotype of immature macrophages or myeloid progenitors has not been studied. Alternatively, differences in the lung microenvironment resulting from specific signaling pathways activated in non-immune cells during the distinct stages of development (e.g. saccular versus alveolar) may represent an additional driver of immune cell phenotype. Comparison of immune subsets in the developing mouse and human lung may provide additional clarity as to the central factors regulating diversity, given that the mouse is in the saccular stage and the human in the alveolar stage immediately prior to birth.

In addition, whether alterations in immune cell diversity may underlie the origin of pediatric lung disease remains to be determined. Many of the Mac clusters express genes with seemingly contrasting functions, highlighting a key role for these populations in the fine-tuned regulation of complex biological process such as inflammation and angiogenesis. Stimuli that disrupt this careful balance, might have profound pathologic effects and contribute to diseases such as BPD, pulmonary hypertension and asthma. For example, in the mouse, inflammatory activation of embryonic macrophages disrupts lung branching (*Blackwell et al., 2011*), and abnormal activation of macrophages is observed in premature infants that develop BPD (*for the Eunice Kennedy Shriver National Institute of Child Health and Human Development Neonatal Research Network et al., 2009*). The detrimental effects of prenatal macrophage activation may result both from the induction of injurious cytokines and chemokines, but also the loss of beneficial homeostatic functions. Further, alterations in the changes in specific macrophage abundance at key points in development may represent an additional origin of specific lung diseases. For example, the transient Mac II population expresses molecules that influence lung and vessel remodeling, including *Tgm2*, a hypoxia induced protein-crosslinker that promotes vascular smooth muscle proliferation and vascular remodeling in experimental PH (*Penumatsa et al., 2017*). Whether an inappropriate persistence of the Mac II cells would be associated with pathologic lung and vascular remodeling is not known. Similarly, the late Mac V subpopulation expresses *Fcgr4*, encoding an Fc receptor capable of binding IgE, suggesting a potential role in modulating airway inflammation and response to inhaled antigen. The alveolar macrophages (Mac III) rapidly increase from E18.5 to P21. Given that resident alveolar macrophages are anti-inflammatory and mitigate the airway hyperresponsiveness that characterizes asthma, a perinatal insult that decreases either Mac V or Mac III abundance could possess long term, pro-inflammatory effects. Further investigation into the effect of injurious stimuli on the phenotype, abundance and localization of these discrete immune cell populations is likely to provide deeper insight into the mechanisms underlying pediatric and adult lung disease.

In summary, these data highlight the marked increase in immune cell diversity after birth, with a developmental plasticity that provides distinct immune populations to fill specific roles in tissue and vascular remodeling, immunoregulation, and bacterial and viral pathogen defense. Injuries to the developing, immature lung can have profoundly untoward and life-long consequences as a significant component of lung parenchymal and vascular development occurs during late pregnancy and the first few years of postnatal life. Many of these injurious stimuli including acute infection, hyperoxia, and corticosteroids are known to have significant effects on immune cell phenotype and function. Therefore, our data provide a detailed framework that enables a more complete understanding of how disruptions of immune cell phenotype may contribute to altered lung development, both through the induction of pathologic, pro-inflammatory signaling as well as the suppression of essential homeostatic functions. Further, a deep understanding of the diversity of immune cell functions during this important window of postnatal development, and how specific immune cell phenotypes are regulated could allow for the application of immunomodulatory therapies as a novel strategy to preserve or enhance lung development in infants and young children.

# Materials and methods

## Key resources table

| Reagent type (species) or resource | Designation | Source or reference | Identifiers | Additional information |
|---|---|---|---|---|
| Strain, strain background (*M. musculus*) | C57BL/6J | Charles Rivers Laboratories | | |
| Antibody | anti-CD3e (Hamster monoclonal) | BD Biosciences | Cat# 553062; RRID:AB_394595 | FACS (1:100) |
| Antibody | anti-CD4 (Rat monoclonal) | BD Biosciences | Cat# 552051; RRID:AB_394331 | FACS (1:100) |
| Antibody | anti- CD8a (Rat monoclonal) | BioLegend | Cat# 100721; RRID:AB_312760 | FACS (1:100) |
| Antibody | anti-Tbet (Mouse monoclonal) | BioLegend | Cat# 644808; RRID:AB_15955479 | FACS (1:100) |
| Antibody | anti-GATA3 (Mouse monoclonal) | BD Biosciences | Cat# 563510; RRID:AB_2738248 | FACS (1:100) |
| Antibody | anti-Foxp3 (Rat monoclonal) | eBioscience | Cat# 12-5773-82; RRID:AB_465936 | FACS (1:100) |
| Antibody | anti-Rorγt (Mouse monoclonal) | BD Biosciences | Cat# 562607; RRID:AB_11153137 | FACS (1:100) |
| Antibody | anti-IFNγ (Rat monoclonal) | BioLegend | Cat# 505829; RRID:AB_10897937 | FACS (1:100) |
| Antibody | anti-IL-4 (Rat monoclonal) | BD Biosciences | Cat# 562045; RRID:AB_10895799 | FACS (1:100) |
| Antibody | IL-10 (Rat monoclonal) | BioLegend | Cat# 505027; RRID:AB_2561522 | FACS (1:100) |
| Antibody | IL-17 (Rat monoclonal) | Miltenyi Biotec | Cat# 130-095-732; RRID:AB_10828821 | FACS (1:100) |
| Commercial assay or kit | FISH probe: anti-mouse Plac8 | Advanced Cell Diagnostics (ADC) | Cat# 532701 | RNAscope: 50 μl per slide |
| Commercial assay or kit | FISH probe: anti-mouse Dab2 | ADC | Cat# 558131-C2 | RNAscope: (1 μl per 50 μl) |
| Commercial assay or kit | FISH probe: anti-mouse Cd68 | ADC | Cat# 316611-C2 | RNAscope: (1 μl per 50 μl) |
| Commercial assay or kit | FISH probe: anti-mouse-Cd68 | ADC | Cat# 316611-C3 | RNAscope: (1 μl per 50 μl) |
| Commercial assay or kit | FISH probe: anti-mouse-Cd68 | ADC | Cat# 316611 | RNAscope: 50 μl per slide |
| Commercial assay or kit | FISH probe: anti-mouse-Cdh5 | ADC | Cat# 312531-C2 | RNAscope: (1 μl per 50 μl) |
| Commercial assay or kit | FISH probe: anti-mouse-Mki67 | ADC | Cat# 416771-C2 | RNAscope: (1 μl per 50 μl) |
| Commercial assay or kit | FISH probe: anti-mouse-Car4 | ADC | Cat# 468421-C2 | RNAscope: (1 μl per 50 μl) |
| Commercial assay or kit | FISH probe: anti-mouse-Gal | ADC | Cat# 400961-C2 | RNAscope: (1 μl per 50 μl) |
| Commercial assay or kit | FISH probe: anti-mouse-C1qa | ADC | Cat# 441221 | RNAscope: 50 μl per slide |
| Commercial assay or kit | FISH probe: anti-mouse-Epcam | ADC | Cat# 418151-C2 | RNAscope: (1 μl per 50 μl) |
| Commercial assay or kit | FISH probe: anti-mouse-Itgax | ADC | Cat# 311501-C3 | RNAscope: (1 μl per 50 μl) |
| Commercial assay or kit | FISH probe: anti-mouse-Ifitm6 | ADC | Cat# 511321-C3 | RNAscope: (1 μl per 50 μl) |

*Continued on next page*

*Continued*

| Reagent type (species) or resource | Designation | Source or reference | Identifiers | Additional information |
|---|---|---|---|---|
| Commercial assay or kit | FISH probe: anti-mouse-Lair1 | ADC | Cat# 509151 | RNAscope: 50 µl per slide |
| Commercial assay or kit | FISH probe: anti-mouse-Ccr2 | ADC | Cat# 501681 | RNAscope: 50 µl per slide |

## Mouse lung cell isolation

C57BL/6 mice were obtained from Charles River Laboratories. For studies using E18.5, P1, and P7 murine lungs, pregnant dams were purchased, and pups aged prior to lung isolation. At E18.5, dam was asphyxiated with $CO_2$ and pups extracted. At P1, P7, and P21 pups were euthanized with euthanasia solution (Vedco Inc). Genetic sex of mice at developmental stages E18.5 and P1 was determined by performing PCR amplification of the Y chromosome gene Sry. P7 and P21 mice were sexed through identification of a pigment spot on the scrotum of male mice (*Wolterink-Donselaar et al., 2009*). For all timepoints, female and male mice were randomly selected for the studies. For all timepoints, except E18.5, the pulmonary circulation was perfused with ice cold heparin in 1x PBS until the circulation was cleared of blood. Lungs were minced and digested with Liberase (Sigma Aldrich) in RPMI for 15 (E18.5, P1, and P7) or 30 (P21) minutes at 37C, 200 rpm. Lungs were manually triturated and 5% fetal bovine serum (FBS) in 1x PBS was used to quench liberase solution. Red blood cells were lysed with 1x RBC lysis buffer (Invitrogen) as indicated by the manufacturer and total lung cells counted on Biorad cell counter (BioRad). Protocols for the murine studies adhered to American Physiological Society/US National Institutes of Health guidelines for humane use of animals for research and were prospectively approved by the Institutional Animal Care and Use Committee at Stanford (APLAC #19087).

## Immunostaining and fluorescence-activated cell sorting (FACS) of single cells

Lungs were plated at $1 \times 10^6$ cells per well and stained with Fc block (CD16/32, 1:100, Tonbo Biosciences) for 30 min on ice. Cells were surface stained with the endothelial marker CD31 (1:100, clone: MEC3.1, eBiosciences), epithelial marker Epcam (1:100, clone: CD326, eBiosciences), and immune marker CD45 (1:100, clone: F11, eBiosciences) for 30 min on ice. The live/dead dye, Sytox Blue (Invitrogen), was added to cells and incubated for 3 min prior to sorting into 384-well plates (Bio-Rad Laboratories, Inc) prefilled with lysis buffer using the Sony LE-SH800 cell sorter (Sony Biotechnology Inc), a 100 µm sorting chip (Catalog number: LE-C3110) and ultra-purity mode. Single color controls were used to perform fluorescence compensation and generate sorting gates. 384-well plates containing single cells were spun down, immediately placed on dry ice and stored at −80C.

## cDNA library generation using Smart-Seq2

Complementary DNA from sorted cells was reverse transcribed and amplified using the Smart-Seq2 protocol on 384-well plates as previously described (*Zanini et al., 2018*; *Picelli et al., 2013*). Concentration of cDNA was quantified using picogreen (Life technology corp.) to ensure adequate cDNA amplification. In preparation for library generation, cDNA was normalized to 0.4 ng/uL. Tagmentation and barcoding of cDNA was prepared using in-house Tn5 transposase and custom, double barcoded indices (*Tabula Muris Consortium et al., 2018*). Library fragment concentration and purity were quantified by Agilent bioanalyzer. Libraries were pooled and sequenced on Illumina NovaSeq 6000 with $2 \times 100$ base kits and at a depth of around 1 million read pairs per cell.

## Data analysis and availability

Sequencing reads were mapped against the mouse genome (GRCm38) using STAR aligner (*Dobin et al., 2013*) and gene were counted using HTSeq (*Anders et al., 2015*). FZ has been the main maintainer of HTSeq for several years. To coordinate mapping and counting on Stanford's high-performance computing cluster, snakemake was used (*Koster and Rahmann, 2012*). Gene expression count tables were converted into loom objects (https://linnarssonlab.org/loompy/) and

cells with less than 50,000 uniquely mapped reads or less than 400 genes per cell were discarded. Of the 4199 immune cells that were filtered, an additional 147 were removed as suspected doublets. Doublets were removed in two ways. First, clusters that expressed epithelial, endothelial or mesenchymal genes were excluded. Clusters showing joint expression of mutually exclusive cell type markers (e.g. *Epcam*, *Cdh5*, *Col6a2*) with *Cd45* were manually excluded as well. Counts for the remaining 4052 cells were normalized to counts per million reads. For t-distributed stochastic embedding (t-SNE) (*van der Maaten and Hinton, 2008*), 500 features were selected that had a high Fano factor in most mice, and the restricted count matrix was log-transformed with a pseudocount of 0.1 and projected onto the top 25 principal components using scikit-learn (*Pedregosa et al., 2011*). Unsupervised clustering was performed using Leiden (C++/Python implementation) (*Traag et al., 2019*). Singlet (https://github.com/iosonofabio/singlet) and for specific analyses, custom Python three scripts were used: the latter available at https://github.com/iosonofabio/lung_neonatal_immune/ (*Domingo-Gonzalez et al., 2020*; copy archived at https://github.com/elifesciences-publications/lung_neonatal_immune). Pathway analysis on the differentially expressed genes from the five macrophage clusters was performed on each the top 100 most differentially expressed genes for each cluster in a one-vs-rest comparison and evaluated the gene set enrichment via Metascape (*Zhou et al., 2019*). The most enriched pathways against a permutation test are shown ordered by significance from top to bottom (negative log of the P-value). Pseudotime was determined with Scanpy: https://scanpy.readthedocs.io/en/stable/api/scanpy.tl.dpt.html?highlight=pseudotime#scanpy.tl.dpt. Local states were computed by interpolating pseudotime as a scalar potential along a grid in the embedding restricted to areas populated by cells and subsequently identifying local pseudotime maxima. The 2D gradient of the potential was computed to determine the vector field (arrows) shown in the figure panel. Because this approach assumes the existence of a unique pseudotime potential, it cannot account for a possible nonzero curl in the vector field (e.g. a self-renewing embryonic population). To address this simplification, RNA velocity was performed (*La Manno et al., 2018*) using scVelo, which explicitly models curly developmental trajectories (https://www.biorxiv.org/content/10.1101/820936v1). To evaluate the global patterns of genes demonstrating low or noisy expression, we calculated a simmetrized k-nearest neighbor similarity graph in PC space and smoothed the count matrix twice over the graph (i.e. reaching second-order neighbors). This approach is similar in spirit but conceptually simpler than MAGIC and other scRNA-Seq imputation algorithms (*van Dijk et al., 2018*). It also has the advantage of working on the actual PC space and knn graph that were used for the Leiden clustering and t-SNE embedding of the cells, ensuring a self-consistent representation of the data across the analysis. T Cell receptors were assembled using TraCeR (*Stubbington et al., 2016*) using the default parameters of the Singularity image. B cell receptors were assembled using BraCeR (*Lindeman et al., 2018*) with the parameter – IGH_networks, which agreed with our in-house pipeline consisting of Basic (*Canzar et al., 2017*) and Change-O (*Gupta et al., 2015*). Northstar was used to compare our data with Tabula Muris (*Tabula Muris Consortium et al., 2018*; *Zanini et al., 2020*, https://www.biorxiv.org/content/10.1101/820928v2), while manual merging and embedding was performed to compare our data with *Schyns et al., 2019*. Raw fastq files, count tables, and metadata are available on NCBI's Gene Expression Omnibus (GEO) website (GSE147668), and the gene count and metadata tables are also available on FigShare at https://figshare.com/articles/Diverse_homeostatic_and_immunomodulatory_roles_of_immune_cells_in_the_developing_mouse_lung_revealed_at_single_cell_resolution/12043365.

## In-situ validation using RNAscope and immunofluorescence (IF)

Embryonic and post-natal mice were euthanized as described above. Female and male mice were randomly selected from the litter, and at least two litters were used to source the lung tissue for all validation studies. E18.5 lungs were immediately placed in 10% neutral buffered formalin following dissection. P1, P7, and P21 murine lungs were perfused as described above, and P7 and P21 lungs inflated with 2% low melting agarose (LMT) in 1xPBS and placed in 10% neutral buffered formalin. Following 20 hr incubation at 4C, fixed lungs were washed twice in 1xPBS and placed in 70% ethanol for paraffin-embedding. In situ validation of genes identified by single cell RNA-seq was performed using the RNAscope Multiplex Fluorescent v2 Assay kit (Advanced Cell Diagnostics) and according to the manufacturer's protocol. Formalin-fixed paraffin-embedded (FFPE) lung sections (5 µm) were used within a day of sectioning for optimal results. Nuclei were counterstained with DAPI (Life

Technology Corp.) and extracellular matrix proteins stained with hydrazide (*Clifford et al., 2011*). Opal dyes (Akoya Biosciences) were used for signal amplification as directed by the manufacturer. Images were captured with Zeiss LSM 780 and Zeiss LSM 880 confocal microscopes, using 405 nm, 488 nm, 560 nm and 633 nm excitation lasers. For scanning tissue, each image frame was set as 1024 × 1024 and pinhole 1AiryUnit (AU). For providing Z-stack confocal images, the Z-stack panel was used to set z-boundary and optimal intervals, and images with maximum intensity were processed by merging Z-stacks images. For all both merged signal and split channels were collected.

### Quantification of RNA scope images

For select RNA scope experiments, the number and location of specific cells were quantified. Adjustments to the brightness and contrast of an image to increase the clarity of the signal was always applied to the entire image and never to separate areas of the image. For the determination of the percentage of $Cd68^+$ cells located in the perivascular versus the perivascular space, all stitch-and-tile images containing large areas of parenchyma and at least on small vessel (20–30 µm) were included in the quantification. For all quantification analysis, a cell was deemed positive if it contained >1 puncta per cell for a specific gene. All studies were performed >2 times, with a total of n = 8 E18.5, n = 6 P1, n = 4 P7, and n = 4 P21 mice used for the FISH studies.

### Intracellular flow cytometry

P1 and P21 male and female murine lungs were isolated as described above. Cells were blocked for 30 min with CD16/CD32 (Tonbo Biosciences). For intracellular analyses, cells were stimulated with PMA (50 ng/ml, Sigma-Aldrich), ionomycin (750 ng/mL, Sigma-Aldrich), and GolgiStop (BD Biosciences) for 5 hr, and then surface stained with fluorochrome-conjugated antibodies for 30 min: CD3 (clone: 145–2 C11, BD Biosciences), CD4 (RM4-5, BD Biosciences), and CD8a (clone: 53–6.7, Biolegend). Cells were then permeabilized with FoxP3 Fixation/Permeabilization Kit (BD Biosciences) as indicated by the manufacturer, and stained for TBET (clone: 4B10, Biolegend), GATA3 (clone: L50-823, BD Biosciences), FOXP3 (clone: FJK-16s, eBioscience), RORγt (clone: Q31-378, BD Biosciences), IFNγ (clone: XMG1.2, Biolegend), IL-4 (clone: 11B11, BD Biosciences), IL-10 (clone: JES5-16E3, Biolegend), and IL-17 (clone: TC11-18H10, Miltenyi Biotec) for 30 min. Cells were read using an LSRII flow cytometer using FACSDiva software. Flow data was analyzed using FlowJo (Tree Star Inc). Flow cytometry analysis for this project was done on instruments in the Stanford Shared FACS Facility.

### Statistical analyses

To identify differentially expressed genes within cell populations, Kolmogorov Smirnov tests on the distributions of gene expression were performed on all genes, and the genes with the largest test statistic (lowest P value) were chosen as the top differentially expressed genes. For the flow cytometry studies of lung lymphocytes, 7–8 mice were in each group. Data are presented as mean ± SD. Differences between groups was determined by Student's t-test with a P value of $\leq 0.05$ considered statistically significant.

## Acknowledgements

We thank Sai Saroja Kolluru (Stanford University) for assistance with library submission to the Chan Zuckerberg Biohub, Yuan Xue (Stanford University) for assistance with the initial single cell RNA-seq data acquisition, Astrid Gillich for technical support with the RNAscope experiments, and Maya Kumar for providing hydrazide. We thank Henry Hampton for his input and fruitful discussions. We also thank the Stanford Shared FACS Facility, Lisa Nichols, Meredith Weglarz, and Tim Knaak for assistance with the flow cytometry instrumentation and antibody panel design. Flow cytometry data was collected on an instrument in the Stanford Shared FACS Facility obtained using NIH S10 Shared Instrument Grant (S10RR027431-01). This work was supported by National Institutes of Health grants HL122918 (CMA), HD092316 (CMA, DNC), the Stanford Maternal Child Health Institute Tashia and John Morgridge Faculty Scholar Award (CMA), the Stanford Center of Excellence in Pulmonary Biology (DNC), Bill and Melinda Gates Foundation (SRQ), and the Chan Zuckerberg Biohub (DNC. and SRQ). MAS is supported by the NSF-GRFP.

## Additional information

### Funding

| Funder | Grant reference number | Author |
| --- | --- | --- |
| National Institutes of Health | HL122918 | Cristina M Alvira |
| National Institutes of Health | HD092316 | David N Cornfield<br>Cristina M Alvira |

The funders had no role in study design, data collection and interpretation, or the decision to submit the work for publication.

### Author contributions

Racquel Domingo-Gonzalez, Data curation, Formal analysis, Validation, Investigation, Methodology, Writing - original draft, Writing - review and editing; Fabio Zanini, Conceptualization, Resources, Data curation, Formal analysis, Investigation, Methodology, Writing - original draft, Writing - review and editing; Xibing Che, Data curation, Investigation, Methodology, Writing - review and editing; Min Liu, Investigation, Writing - review and editing; Robert C Jones, Conceptualization, Investigation, Methodology, Writing - review and editing; Michael A Swift, Formal analysis, Investigation, Methodology, Writing - review and editing; Stephen R Quake, Conceptualization, Resources, Formal analysis, Supervision, Writing - review and editing; David N Cornfield, Conceptualization, Resources, Formal analysis, Supervision, Funding acquisition, Investigation, Methodology, Writing - review and editing; Cristina M Alvira, Conceptualization, Resources, Data curation, Formal analysis, Supervision, Funding acquisition, Validation, Investigation, Visualization, Methodology, Writing - original draft, Project administration, Writing - review and editing

### Author ORCIDs

Fabio Zanini (iD) http://orcid.org/0000-0001-7097-8539
Robert C Jones (iD) http://orcid.org/0000-0001-7235-9854
Stephen R Quake (iD) https://orcid.org/0000-0002-1613-0809
Cristina M Alvira (iD) https://orcid.org/0000-0002-6921-0001

### Ethics

Animal experimentation: This study was performed in strict accordance with the recommendations in the Guide for the Care and Use of Laboratory Animals of the National Institutes of Health. All of the animals were handled according to approved institutional animal care and use committee (IACUC) protocols (#19087) of Stanford University School of Medicine.

### Decision letter and Author response

Decision letter https://doi.org/10.7554/eLife.56890.sa1
Author response https://doi.org/10.7554/eLife.56890.sa2

## Additional files

### Supplementary files

• Transparent reporting form

### Data availability

Sequencing data have been deposited in GEO under accession code GSE147668. Gene count and metadata tables are also available on FigShare at https://figshare.com/articles/Diverse_homeostatic_and_immunomodulatory _roles_of_immune_cells_in_the_developing_mouse_lung_revealed_at_single_cell_resolution/12043365.

The following dataset was generated:

| Author(s) | Year | Dataset title | Dataset URL | Database and Identifier |
|---|---|---|---|---|
| Domingo-Gonzalez R, ZaniniF, Che X, Liu M, Jones RC, Swift MA, Quake SR, Cornfield DN, Alvira CM | 2020 | Diverse homeostatic and immunomodulatory roles of immune cells in the developing mouse lung revealed at single cell resolution | https://www.ncbi.nlm.nih.gov/geo/query/acc.cgi?acc=GSE1476668 | NCBI Gene Expression Omnibus, GSE147668 |

The following previously published datasets were used:

| Author(s) | Year | Dataset title | Dataset URL | Database and Identifier |
|---|---|---|---|---|
| Schyns J, Bai Q, Ruscitti C, Rader-mecker C, De Schepper S, Chakarov S, Pirottin D, Ginhoux F, Boeckxstaens G, Bureau F, Marichal T | 2019 | scRNA-seq analysis of lung CD64-expressing mononuclear cells, patrolling and classical monocytes from steady-state C57BL/6J mice | https://www.ebi.ac.uk/arrayexpress/experiments/E-MTAB-7678/ | ArrayExpress, 10.1038/s41467-019-11843-0 |
| Tabula Muris Consortium | 2018 | Tabula Muris: Transcriptomic characterization of 20 organs and tissues from Mus musculus at single cell resolution: Single-cell RNA-seq data from Smart-seq2 sequencing of FACS sorted cells (v2) | https://figshare.com/projects/Tabula_Muris_Transcriptomic_characterization_of_20_organs_and_tissues_from_Mus_musculus_at_single_cell_resolution/27733 | FigShare, 10.1038/s41586-018-0590-4 |

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
