## [Decision Letter]

**Acceptance summary:**

This work provides a comprehensive bioinformatic view of the immune cell landscape of the developing mouse lung at an important biological transition from in utero to the neonatal air-breathing period. The data are coupled with well-quantified spatial analysis of some of the most dynamic cell types identified in the analysis. They are also presented clearly in the context of analogous recent studies and provide a biological overview of immune cell evolution in mouse lung perinatal development.

**Decision letter after peer review:**

Thank you for submitting your article "Diverse homeostatic and immunomodulatory roles of immune cells in the developing mouse lung at single cell resolution" for consideration by *eLife*. Your article has been reviewed by three peer reviewers, one of whom is a member of our Board of Reviewing Editors, and the evaluation has been overseen by Tadatsugu Taniguchi as the Senior Editor. The reviewers have opted to remain anonymous.

The reviewers have discussed the reviews with one another and the Reviewing Editor has drafted this decision to help you prepare a revised submission.

Summary:

The present study by Domingo-Gonzales et al. is a comprehensive bioinformatic approach, in which they investigate immune cells in the lung perinatally by characterizing changes in composition, localization, and function before birth and through the first three weeks of life. To this end, they combined a transcriptomic analysis (single cell RNA-seq) with fluorescent in situ hybridization. They clearly identified and characterized the localization and dynamics of five macrophage/monocyte populations with very distinct functions in murine lungs. Moreover, they provide strong data on the abundance and distribution of other important immune cells (e.g. dendritic cells) that are central in lung developmental processes, but also in lung diseases. Overall the manuscript is very well written addressing an highly important and disease-relevant topic that has not been elucidated to this extent yet.

Essential revisions:

These can all be done without additional experiments.

1) The quality of the spatial information describing the location of the macrophages/monocytes needs to be improved. A detailed quantification of the in situ hybridisation images is required for clarifying the distribution of perivascular macrophage/monocytes in the tissues e.g. in Figure 2E, F all Cd68^+^ cells should be scored as Dab2^+^, Plac8^+^, both, or neither. In Figure 3 the % macrophages around the vessels versus scattered should be given, the relative % of type 1 versus IV around the vessels and the % of co-labelling with *Mki67*. Moreover, clarification of the structures seen in Figure 3 would be helpful – are the macrophages arranged in intermittent circles around the diameter of the vessels? Are they forming lines along their length (if so, from where to where)? If possible, an antibody-based immunofluorescence approach to improve the cellular resolution of the image data would be desirable.

2) The data in Figure 4A and the respective tables are very interesting, providing new insight in the functional role of the macrophages. Pathway analysis using these respective genes in the supplementary tables would give more information of the pathways and functional regulation of the individual macrophage types.

3) Figure 3 and Figure 4 suggest that some macrophage population, e.g. Mac I, might regulate and promote angiogenesis in the lung. These findings coupled with Figure 4 showing genes related to angiogenesis let me query if anti-angiogenic factors might be produced by specific macrophages. That might be of interest in regulating alveolarization, but also in the pathogenesis of lung diseases (PAH or BPD).

4) Did the authors relate macrophages or other immune cells, such as dendritic cells or B cells, to the regulation of stemness or lung progenitor cells? They are crucial in the process of alveolarization.

5) We recommend including Fc receptor data in the main manuscript. They are important in disease development and often a therapeutic target.

6) How do the present mouse data relate to human lung? At birth, murine lungs are in the saccular stage, whereas human lungs are already at the alveolar stage. What might be the main driver of macrophage differentiation: oxygen? mechanical forces? That could be further addressed in the Discussion.

7) Discussion: It would be interesting to discuss more in detail how this precise regulation of immune cell differentiation and distribution in the lung during development could underlie the origins of lung disease. For example, there are data showing that BPD is related to activation of macrophages. Could asthma originate from adverse effects on dendritic and T cells during a critical phase of differentiation? That could be discussed.

8) To make this manuscript more useful to the community further details need to be provided about the bioinformatic analysis:

a) The authors should state exactly how many cells in total were profiled and how many cells passed QC. It will be useful as part of QC metric to see the number of genes per cell state identified.

b) The link to scripts, gene count and metadata table provided in the manuscript was inactive.

c) The authors should state what is actually displayed as gene expression for all figures – is this expression value on log scale? What does min, mod, high e.g. Figure 4A refer to quantitatively?

d) Smart-seq2 is less likely to have doublet contamination but it is not clear if doublet detection and removal during analysis was attempted.

e) Was batch correction methods e.g. Harmony implemented? – this would be helpful to assess if findings were impacted by batch effects.

f) The observation of Mac I that encircles blood vessels prenatally and disappears after birth is interesting. However, the claims based on using inferred trajectory analysis of scRNA-seq data to suggest Mac I transition into Mac II and III needs to be more cautiously made as definitive lineage tracing or fate-mapping experiments were not done to validate the trajectory inference.

g) What pseudotime trajectory method was used?

9) DC nomenclature may be confusing as DC1, DC2 and DC3 are recognised subsets of conventional DCs. DCII described by authors are monocyte-derived DC and DCIII identity is unclear. This should be clarified in the text.

10) Some integration of this dataset with other existing datasets from embryonic, perinatal and adult immune cells would permit harmonization of the cell states described by the various manuscripts and enhancing the value of this manuscript from a biological advancement and resource value of the manuscript.

---

## [Author Response]

Essential revisions:These can all be done without additional experiments.1) The quality of the spatial information describing the location of the macrophages/monocytes needs to be improved. A detailed quantification of the in situ hybridisation images is required for clarifying the distribution of perivascular macrophage/monocytes in the tissues e.g. in Figure 2E, F all Cd68^+^ cells should be scored as Dab2^+^, Plac8^+^, both, or neither. In Figure 3 the % macrophages around the vessels versus scattered should be given, the relative % of type 1 versus IV around the vessels and the % of co-labelling with Mki67. Moreover, clarification of the structures seen in Figure 3 would be helpful – are the macrophages arranged in intermittent circles around the diameter of the vessels? Are they forming lines along their length (if so, from where to where)? If possible, an antibody-based immunofluorescence approach to improve the cellular resolution of the image data would be desirable.

We very much appreciate this point from the reviewers. We have quantified the in situ hybridization images to provide the: (i) percent Dab2^+^ vs. Plac8^+^ cells at E18.5 and P7; (ii) the percentage of perivascular vs. parenchymal macrophages at E18.5, (iii) the % proliferating macrophages; and (iv) the % of Mac I vs. Mac IV found around vessels at E18.5. These data are now included in Figure 2E and Figure 3D, E, and G. Whether the macrophages are forming lines along the length of the vessels is an intriguing and important question that is difficult for us to definitively answer with our current images. Although some images that capture both the cross-section and a longitudinal cut of the vessel suggest that the macrophages may also be present along the length of the vessel, we cannot claim this to be true with confidence. We now acknowledge the limitation in the Discussion and state that antibody-based immunofluorescence followed by Z-stack imaging of thick tissue sections would be required to definitively address this important question.

2) The data in Figure 4A and the respective tables are very interesting, providing new insight in the functional role of the macrophages. Pathway analysis using these respective genes in the supplementary tables would give more information of the pathways and functional regulation of the individual macrophage types.

We agree. We have performed pathway analysis using the DEGs from the five macrophage/monocyte populations and have included these data as Figure 4—figure supplement 1, and top pathways for each population reported in the Results. Given that many top DEGs have published functions that are not included in the pathway analysis, we also continue to highlight individual genes of interest with the relevant citations.

3) Figure 3 and Figure 4 suggest that some macrophage population, e.g. Mac I, might regulate and promote angiogenesis in the lung. These findings coupled with Figure 4 showing genes related to angiogenesis let me query if anti-angiogenic factors might be produced by specific macrophages. That might be of interest in regulating alveolarization, but also in the pathogenesis of lung diseases (PAH or BPD).

We agree with the reviewer that the production of anti-angiogenic factors may be important not only for understanding alveolarization but also relevant to lung diseases such as PAH and BPD. We now also highlight some DEGs that are reported to have anti-angiogenic functions, including *Serpine1*, and *Pf4* (subsection “Distinct transcriptional profiles and spatial distribution suggest specific physiologic functions for discrete macrophage populations”, third and fourth paragraphs).

4) Did the authors relate macrophages or other immune cells, such as dendritic cells or B cells, to the regulation of stemness or lung progenitor cells? They are crucial in the process of alveolarization.

Although it was not our initial focus, we agree that this is a very important point given the role of lung progenitor cells in all stages of lung development. We also looked at the expression of secreted growth factors and other effectors of key pathways that regulate lung progenitor cells including the *Wnt*, Shh, Tgf-β, and BMP pathways. Of note, the Mac II and Mac IV clusters expressed secreted molecules that regulate these pathways. The Mac II cells express both Tgf-β1 and 2, Nestin 1 and 4 and *Hhip*, while the Mac IV cells expressed *Fgf10*, *Bmp4*, *Wnt2* and *Wnt5*. These data are now detailed in the Results (subsection “Distinct transcriptional profiles and spatial distribution suggest specific physiologic functions for discrete macrophage populations”, second and fourth paragraphs) and included as Figure 4—figure supplement 2.

5) We recommend including Fc receptor data in the main manuscript. They are important in disease development and often a therapeutic target.

Thank you for this recommendation. The Fc receptor data is now included in the main manuscript as Figure 7.

6) How do the present mouse data relate to human lung? At birth, murine lungs are in the saccular stage, whereas human lungs are already at the alveolar stage. What might be the main driver of macrophage differentiation: oxygen? mechanical forces? That could be further addressed in the Discussion.

We thank the reviewers for this important point. An additional paragraph has been added to the Discussion (fifth paragraph) addressing factors that might be driving immune cell diversity in the developing lung and how these data from the mouse might be compared to similar data from human lungs to better understand whether the physiologic changes at birth, as opposed to differences in the lung immune cell niche at the different stages of development might be the predominant driver of diversity.

7) Discussion: It would be interesting to discuss more in detail how this precise regulation of immune cell differentiation and distribution in the lung during development could underlie the origins of lung disease. For example, there are data showing that BPD is related to activation of macrophages. Could asthma originate from adverse effects on dendritic and T cells during a critical phase of differentiation? That could be discussed.

We agree with this excellent point. We have addressed this by including an additional paragraph in the Discussion relating our findings to pediatric lung diseases such as bronchopulmonary dysplasia, pulmonary hypertension and asthma (Discussion, sixth paragraph).

8) To make this manuscript more useful to the community further details need to be provided about the bioinformatic analysis:a) The authors should state exactly how many cells in total were profiled and how many cells passed QC. It will be useful as part of QC metric to see the number of genes per cell state identified.

Thank you for this suggestion. Please see the breakdown of the parameters and QC during data ingestion and filtering as outlined in Author response table 1. Please note that the dataset was sequenced in combination with other studies on the same biological samples. Therefore, the number of immune cells at the second to last step is much smaller than the total number of sequenced cells, which include other cell populations processed from the same plates.

Author response table 1

b) The link to scripts, gene count and metadata table provided in the manuscript was inactive.

We would like to thank the reviewers for identifying this error. The link to the scripts was inactive due to a renaming issue which has been corrected. The metadata and gene counts are available on FigShare and on GEO as stated.

c) The authors should state what is actually displayed as gene expression for all figures – is this expression value on log scale? What does min, mod, high e.g. Figure 4A refer to quantitatively?

Thank you for this request, which we clarified in the revised manuscript. The color scheme of the gene expression tSNE embedding is logarithmic viridis color scales, with a pseudocount of 0.1 counts per million, and normalized to the highest expressing cell. The color scheme of the heatmaps plasma, logarithmic with a pseudocount of 0.1 counts per million, and normalized to the highest expressing population for each gene – i.e. every column will have one or more populations expressing “high”/yellow. The legend refers to 0%, 33%, 67%, and 100% expression (on a log scale) compared to the highest expressing population, i.e. describes equally spaced intervals of expression (on a log scale). These details have been added to the figure legends.

d) Smart-seq2 is less likely to have doublet contamination but it is not clear if doublet detection and removal during analysis was attempted.

Doublets were removed in two ways. First, clusters that expressed epithelial, endothelial or mesenchymal genes were excluded. Clusters showing joint expression of mutually exclusive cell type markers (e.g. *Epcam*, *Cdh5*, *Col6a2*) with *Cd45* were manually excluded as well. 147 out of 4199 cells were removed, which is a slightly conservative filter compared to the expected doublet rate of around 1-2% in 384-well plates on our cell sorter. We have added this information to the Materials and methods (subsection “Data analysis and availability”).

e) Was batch correction methods e.g. Harmony implemented? – this would be helpful to assess if findings were impacted by batch effects.

We purposely did not run batch effect correction methods to avoid eliminating potentially relevant biological signals (e.g. gender and developmental effects). In our experience, such batch correction methods are primarily needed when mixing different technologies. However, during the feature selection step for calculating neighborhood graphs towards clustering and embeddings, we identified over-dispersed features within each mouse and only kept features found in several mice. This enabled us to maintain a straightforward biological interpretation for the counts (reads per million mapped uniquely) while mitigating technical artifacts in the visualizations.

f) The observation of Mac I that encircles blood vessels prenatally and disappears after birth is interesting. However, the claims based on using inferred trajectory analysis of scRNA-seq data to suggest Mac I transition into Mac II and III needs to be more cautiously made as definitive lineage tracing or fate-mapping experiments were not done to validate the trajectory inference.

Thank you for this comment. We completely agree, and we have added an additional sentence to the Results stating that definitive genetic lineage tracing studies are necessary to confirm the transition of the Mac I cells to Mac II and III. We also added RNA velocity to Figure 4G, as an additional method supporting the pseudotime analysis.

g) What pseudotime trajectory method was used?

We apologize for omitting this detail. For the pseudotime analysis we used the default algorithm in Scanpy. This information is included in the Materials and methods. We also added RNA velocity data to the manuscript, and this was performed using scVelo (Bergen et al., 2019, https://doi.org/10.1101/820936).

9) DC nomenclature may be confusing as DC1, DC2 and DC3 are recognised subsets of conventional DCs. DCII described by authors are monocyte-derived DC and DCIII identity is unclear. This should be clarified in the text.

We appreciate this comment and agree that the suspected DC identity was not clear in the prior version. We have examined numerous additional genes associated with cDC1 and cDC2 and find that the prior population we called “DCII” are most consistent with cDC2 rather than monocyte-derived DC. We have included these additional t-SNE plots in Figure 8—figure supplement 1, and renamed our DC populations: cDC1, cDC2, and migratory (mig)-DC.

10) Some integration of this dataset with other existing datasets from embryonic, perinatal and adult immune cells would permit harmonization of the cell states described by the various manuscripts and enhancing the value of this manuscript from a biological advancement and resource value of the manuscript.

Thank you for this excellent suggestion. To this end, we have analyzed our data in concert with two previously published data sets: Schyns et al., 2019, and Tabula Muris, 2018. The results from these harmonized data analyses are included in the manuscript (subsection “Comparison of Mac I-V populations with adult murine lung data sets”), and the data included in a new figure (Figure 6).